# SMARCB1 loss activates patient-specific distal oncogenic enhancers in malignant rhabdoid tumors

Ning Qing Liu [1,2,7], Irene Paassen[3,4,7], Lars Custers[3,4,7], Peter Zeller[4,5,6], Hans Teunissen[1], Dilara Ayyildiz[3,4], Jiayou He[3,4], Juliane Laura Buhl[3,4], Eelco Wieger Hoving [3], Alexander van Oudenaarden [4,5,6], Elzo de Wit [1] ✉ & Jarno Drost [3,4] ✉

Malignant rhabdoid tumor (MRT) is a highly malignant and often lethal childhood cancer. MRTs are genetically defined by bi-allelic inactivating mutations in *SMARCB1*, a member of the BRG1/BRM-associated factors (BAF) chromatin remodeling complex. Mutations in BAF complex members are common in human cancer, yet their contribution to tumorigenesis remains in many cases poorly understood. Here, we study derailed regulatory landscapes as a consequence of *SMARCB1* loss in the context of MRT. Our multi-omics approach on patient-derived MRT organoids reveals a dramatic reshaping of the regulatory landscape upon *SMARCB1* reconstitution. Chromosome con- formation capture experiments subsequently reveal patient-specific looping of distal enhancer regions with the promoter of the *MYC* oncogene. This intertumoral heterogeneity in *MYC* enhancer utilization is also present in patient MRT tissues as shown by combined single-cell RNA-seq and ATAC-seq. We show that loss of *SMARCB1* activates patient-specific epigenetic repro- gramming underlying MRT tumorigenesis.

The BRG1/BRM-associated factors (BAF) chromatin remodeling com- plex (or mammalian SWItch/Sucrose Non-Fermentable (SWI/SNF) complex) is a multiprotein complex with a crucial role in the regulation of transcription via nucleosome positioning. Depending on their pro- tein composition, three main BAF complexes have been identified. The canonical BAF (cBAF) and polybromo-associated BAF (pBAF) com- plexes are defined by, amongst others, the SMARCB1 subunit, which is lacking in the non-canonical (nc)BAF complex[1]. Instead, the ncBAF complex contains the BRD9 and GLTSCR1/L subunits. Genes encoding members of the BAF complex are among the most commonly mutated genes in cancer, occurring in approximately 20-25% of cases[2–4]. Adult malignancies typically harbor hundreds to thousands of mutations, which complicates studying the contribution of mutations in BAF members to tumorigenesis. However, several pediatric malignancies are uniquely defined by mutations in BAF complex members[5]. Amongst those are malignant rhabdoid tumors (MRT), aggressive childhood malignancies that predominantly affect infants. They can occur throughout the body, but most commonly arise in the brain (atypical teratoid/rhabdoid tumors (AT/RT)) and kidney[6]. Despite their aggressiveness, MRTs are genetically stable tumors with a low muta- tional burden. In fact, in the vast majority of MRT cases (95%), bi-allelic inactivation of the BAF complex member *SMARCB1* is the only recur- rent gene alteration, showing that loss of *SMARCB1* is a crucial driver event in MRT development[6,7]. Earlier studies revealed the importance of *SMARCB1* loss to drive MRT formation. For instance, *SMARCB1* loss at a certain time window during murine embryonic development is

[1]Division of Gene Regulation, Netherlands Cancer Institute, Amsterdam, the Netherlands. [2]Department of Hematology, Erasmus Medical Center (MC) Cancer Institute, Rotterdam, the Netherlands. [3]Princess Máxima Center for Pediatric Oncology, Utrecht, the Netherlands. [4]Oncode Institute, Utrecht, the Netherlands. [5]Hubrecht Institute-KNAW, Utrecht, the Netherlands. [6]University Medical Center Utrecht, Utrecht, the Netherlands. [7]These authors contributed equally: Ning Qing Liu, Irene Paassen, Lars Custers. ✉e-mail: e.d.wit@nki.nl; J.Drost@prinsesmaximacentrum.nl

sufficient to initiate MRT formation[8–10]. We recently found that bi-allelic *SMARCB1* inactivating mutations in MRT can be shared with adjacent morphologically normal Schwann cells, suggesting that loss of *SMARCB1* is required but not sufficient for MRT development[11]. No other recurrent genetic alterations have been identified in MRT[6,12–14], suggesting that cell type or cell state-specific epigenetic mechanisms guided by *SMARCB1* loss further drive malignant transformation.

Recent studies have demonstrated that epigenetic reprogramming, which can be caused by mutations in chromatin regulators such as BAF complex members, underpins tumor initiation and progression[15–19]. To identify and functionally study patient-specific epigenomic changes underlying malignant growth, personalized pre-clinical cell models are indispensable. Such models should reflect the epigenetic heterogeneity found between patient tumors. We and others have previously demonstrated that organoids maintain many features of the tissues they were derived from, including their epigenetic profiles, over extended serial passaging[20–23].

Here, we apply a multi-omics approach to patient-derived organoids (PDOs) to define how loss of *SMARCB1* reorganizes chromatin and underpins MRT growth. We find patient-specific enhancer landscapes that are a consequence of a disruption in the balance of BAF complexes. Specifically, we identify patient-specific enhancers in different PDOs that may regulate the expression of the *MYC* oncogene driving MRT growth. The patient-specific derailed activity of these putative enhancers is also observed in patient tumors. Our study shows that intertumoral heterogeneity in enhancer utilization drives oncogene expression and tumorigenesis in MRT.

## Results

### Analysis of chromatin dynamics in MRT PDOs reveals SMARCB1-dependent enhancer regulation

We previously showed that *SMARCB1* loss is required but not sufficient for malignant transformation[11]. We therefore hypothesized that additional epigenetic drivers are required for tumor formation. Reconstitution of the principal genetic driver event of MRT, *SMARCB1* loss, reverts MRT cells to a normalized state[11], suggesting that epigenetic changes that have contributed to malignancy can be overcome (Fig. 1A and Supplementary Fig. 1A). To find these tumor-driving regulatory changes, we lentivirally transduced an MRT PDO model (named P103)[23] with either a Luciferase expression (Control) or a SMARCB1 expression (SMARCB1 + ) plasmid and measured chromatin accessibility by assay for transposase-accessible chromatin using sequencing (ATAC-seq) and BAF chromatin occupancy by chromatin immunoprecipitation sequencing (ChIP-seq) or Cleavage Under Targets & Release Using Nuclease sequencing (CUT&RUN) (Fig. 1A, B and Supplementary Fig. 2A, Supplementary Data 1). Following *SMARCB1* reconstitution, we found 7,941 newly formed open chromatin regions (OCRs) that are enriched for transcription factor motifs from different families such as SMARCC1/2 and AP-1[24] (Fig. 1C, Supplementary Fig. 1B, C). SMARCB1 ChIP-seq revealed that these OCRs are bound by SMARCB1 (cBAF and PBAF) and SS18 (ncBAF and cBAF), indicating cBAF binding at these regions upon reconstitution (Supplementary Fig. 2B). When we performed functional annotation of these OCRs using GREAT, we found that several categories are enriched, mostly related to differentiation and developmental processes (Supplementary Fig. 1D). At the 1,211 OCRs that were lost, an apparent decrease in binding was observed of the ncBAF complex members BRD9 and SS18 (Supplementary Fig. 2B). The only motif that we found to be enriched in these regions was for the insulator protein CTCF (Supplementary Fig. 1C), consistent with previous reports in human embryonic stem cells and mouse embryonic fibroblasts[24,25].

ChIP-seq of CTCF showed that *SMARCB1* reconstitution results in a decrease in CTCF binding at OCRs that are lost in SMARCB1+ cells (Fig. 1C). In the genome, CTCF binding frequently overlaps with binding of the ring-shaped cohesin complex[26]. ChIP-seq of the cohesin

complex subunit RAD21 confirmed that, in addition to decreased CTCF binding, OCRs lost in SMARCB1+ cells show a reduction in binding of the cohesin complex (Fig. 1C). On the other hand, OCRs gained in the SMARCB1+ PDOs showed occupancy of RAD21 (Fig. 1C), but not CTCF. To determine how the enhancer landscape changes upon *SMARCB1* reconstitution, we performed ChIP-seq of the active enhancer mark H3K27ac. As expected, the lost OCRs showed a decrease in H3K27ac levels, while a marked increase in H3K27ac levels was detected at sites that gained accessibility (Fig. 1C). Increase in the active enhancer mark coincided with increased binding of the cohesin complex (Fig. 1C), which is consistent with a subset of cohesin molecules binding to (super) enhancer regions, besides CTCF binding sites[27,28]. These results indicate that the canonical BAF complexes play an important role in suppressing CTCF binding to chromatin and that loss of SMARCB1 leads to increased accessibility at CTCF-bound chromatin as well as ncBAF complex binding to tumor-specific OCRs.

### *SMARCB1* reconstitution reorganizes the chromatin landscape surrounding the *MYC* oncogene

Observing a change in the CTCF and RAD21 binding landscape following *SMARCB1* reconstitution led us to hypothesize that long-distance gene regulation may be affected in these cells. To determine whether any SMARCB1-dependent changes occur in the organization of the genome (3D genome), we performed high-resolution in-situ Hi-C experiments[29] in control and SMARCB1 + P103 MRT PDOs. Despite the differences in CTCF binding to chromatin, we found that chromatin architecture is largely unaffected (Supplementary Fig. 3A–C). 3D genome features such as topologically associated domains (TADs) and A/B compartments showed limited changes upon *SMARCB1* reconstitution (Supplementary Fig. 3B-H). However, we identified 131 and 1,164 loops specific to control and SMARCB1+ PDOs, respectively, with a minimal 1.5-fold change of contact frequency (Fig. 2C, Supplementary Fig. 4A, B). Ranking the most prominently lost loci upon *SMARCB1* reconstitution, we found an interaction between the promoter of the *MYC* oncogene and a -1.1 Mb distal region (Fig. 2A, C). This loop is the sixth most reduced interaction following *SMARCB1* reconstitution, and the top-ranked interaction involving a proto-oncogene. This distal region of the *MYC* promoter is marked by high H3K27ac levels indicative of a super enhancer (Fig. 2 A, B and Supplementary Data 2).

The *MYC* oncogene is of particular interest in the context of MRT, as it was previously demonstrated by us and others that MRTs are defined by a SMARCB1-dependent *MYC* gene expression signature[11,12]. Similarly, we observed a strong reduction in *MYC* mRNA and protein levels as well as deregulation of cell cycle-related genes upon *SMARCB1* reconstitution, indicative of a cell cycle arrest (Fig. 2D and Supplementary Fig. 4D). Moreover, shRNA-induced knockdown of *MYC* in MRT PDOs resulted in a significant reduction of cell proliferation (Fig. 2E and Supplementary Fig. 4E), demonstrating that MYC is required for MRT growth. In addition to a decrease in *MYC* expression, we observed that chromatin accessibility at the super-enhancer, as well as at the *MYC* promoter strongly diminished upon reconstitution of *SMARCB1* (Fig. 2A, B). To explore whether ncBAF complex binding was affected by *SMARCB1* reconstitution, we performed CUT&RUN on BRD9 (ncBAF) and SS18 (cBAF and ncBAF). We found that both BRD9 and SS18 binding at the *MYC* promoter as well as at the 1.1 Mb distal region is dramatically reduced upon *SMARCB1* reconstitution (Fig. 2B). These results indicate that *MYC* expression is at least for a large part dependent on binding of the ncBAF complex. Reconstitution of *SMARCB1* reduces ncBAF complex binding at the *MYC* locus thereby potentially reducing the distal super enhancer with the *MYC* promoter.

### Patient-specific super-enhancers interact with *MYC* in MRT

We then hypothesized that *MYC* distal enhancer looping may be a general mechanism driving MRT oncogenesis. To this end, we

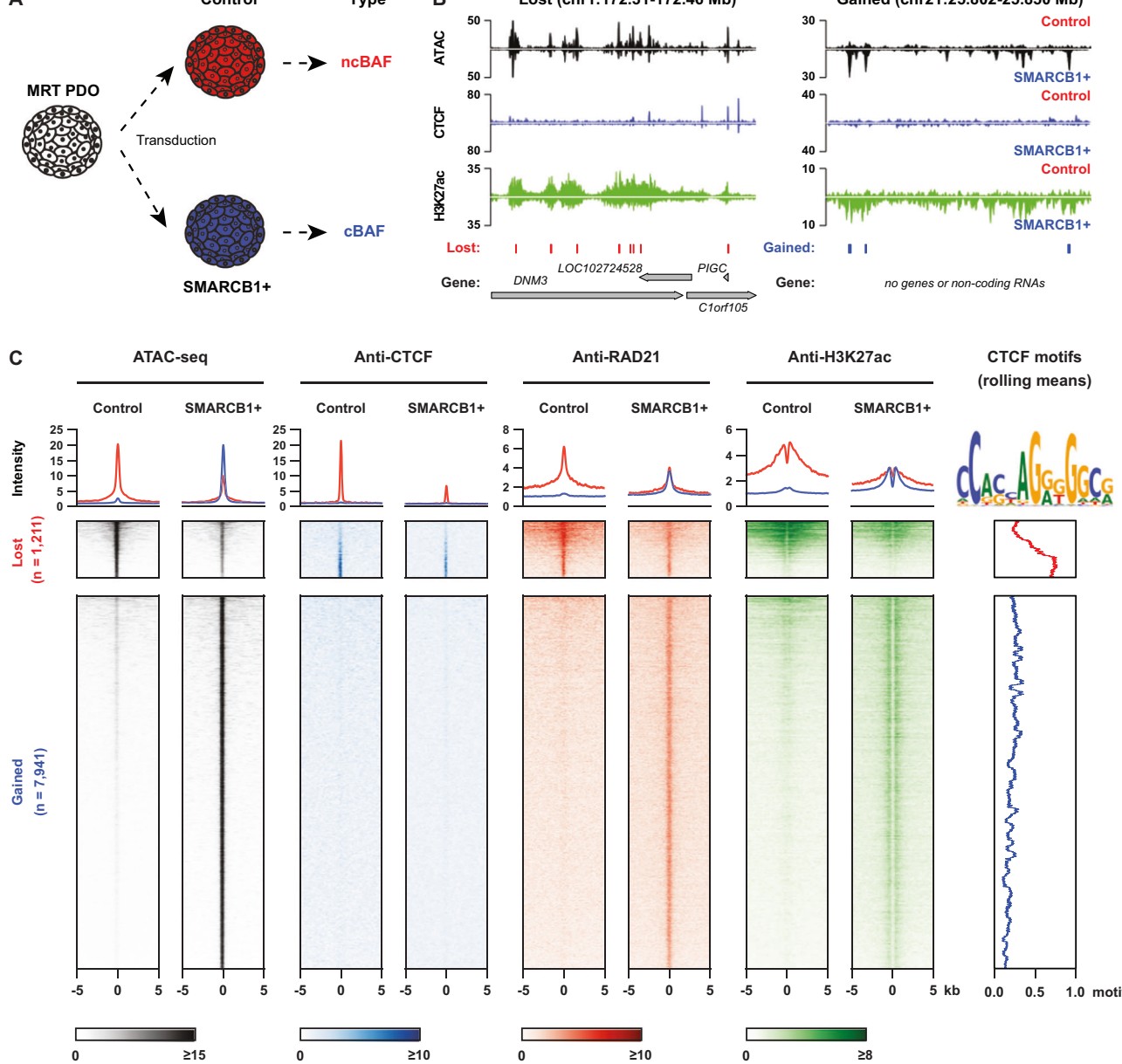

**Fig. 1 | *SMARCB1* reconstitution reshapes open chromatin landscape in the MRT PDO model. A** Schematic overview of the experimental set-up of lentiviral transduction (adapted from Custers et al.[11]); **B** Representative loci that lose (left) or gain (right) chromatin accessibility after SMARCB1 reconstitution (ATAC-seq: *n* = 3, an average of the 3 replicates; ChIP-seq of CTCF and H3K27Ac: *n* = 1). Called lost and gained peaks are indicated below the tracks (FDR cut-off <0.05, FC difference in accessibility > 2); **C** Tornado-plot showing that the lost (control-specific) open chromatin regions are enriched for the binding of CTCF, RAD21 and H3K27ac, while the gained (SMARCB1 + -specific) open chromatin regions are only enriched for RAD21 and H3K27ac (CTCF ChIP *n* = 1, RAD21 ChIP *n* = 1, H3K27Ac ChIP *n* = 1).

performed *SMARCB1* reconstitution in two additional MRT PDOs (P78 and P60). RNA-seq analysis revealed a decrease in *MYC* expression in both these MRT models, as well as deregulation of cell cycle genes confirming the induction of a proliferation arrest (Fig. 2D, E). Next, we performed 4C-seq, a chromosome conformation capture method targeted to a specific genomic site, using the *MYC* promoter as the viewpoint, on these two additional PDOs. Similar to P103, P78 showed a reduction in *MYC* promoter contact frequency with the same -1.1 Mb distal genomic region (Fig. 3A) upon *SMARCB1* reconstitution. Remarkably, 4C-seq analysis in the third PDO model (P60) showed no interaction of the *MYC* promoter with our previously identified genomic region. Instead, 4C-seq uncovered two other chromatin loops that were diminished upon *SMARCB1* reconstitution (Fig. 3A).

To chart the regulatory landscape of the PDO models in the presence and absence of SMARCB1, we performed ATAC-seq on P78 and

P60. When we overlaid the ATAC-seq data with the 4C-seq data, we found that the regions that interact with the *MYC* promoter were highly accessible. The accessibility correlated with the patient-specific *MYC* promoter interaction landscape (Fig. 3B). Upon *SMARCB1* reconstitution, the loss of interactions coincides with a loss of accessibility at these loci (Fig. 3B). The stretches of accessible chromatin in P78 resemble the super-enhancer we identified in P103. Collectively, we refer to these putative super-enhancer regions as Rhabdoid Oncogenic *MYC* Enhancer (RhOME) and number them according to their position in the genome. Notably, in contrast to RhOME2, RhOME1 (*PCAT1*) and RhOME3 (*CCDC26*) were previously identified as *MYC* regulatory regions in other tumor entities[30-36]. Our Hi-C and 4C experiments thus revealed that in different MRT PDOs, distinct distal enhancers interacting with the *MYC* promoter are active (RhOME1-3). The intertumoral specificity of the RhOMEs was further highlighted by the expression of

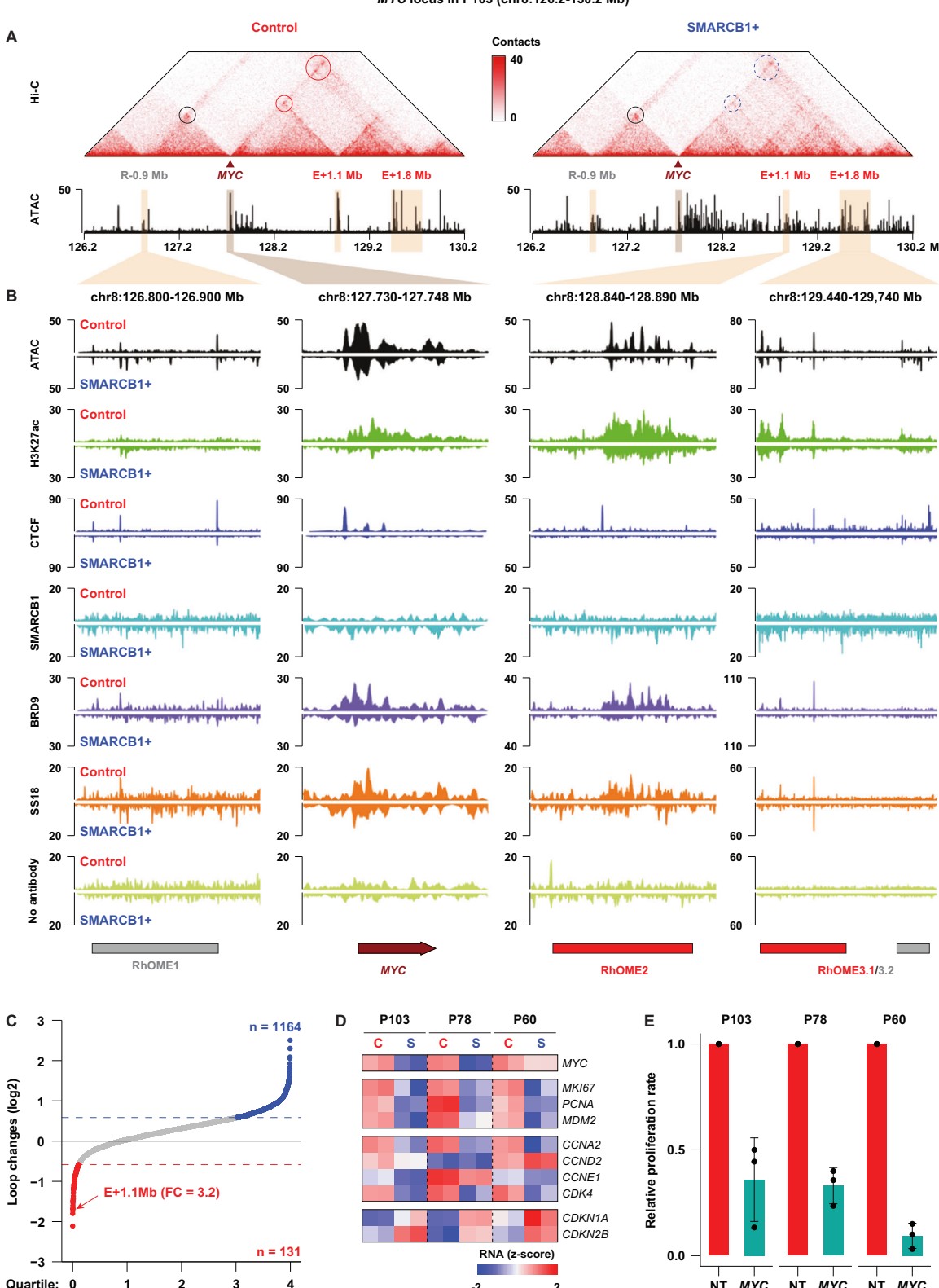

enhancer RNAs (eRNAs), distinctive for active super enhancers[37], specifically in the MRT PDOs in which these enhancers are active and interacting with the *MYC* promoter (Fig. 3C). These results indicate that loss of *SMARCB1* induces an epigenetic state that is characterized by the formation of long-range promoter-enhancer loops that are specific to a PDO model, but heterogeneous between MRT PDOs derived from different donors.

Because MRT PDOs, to a large extent, retain the (epi-)genetic characteristics of the tissues they were derived from[23,38], we used these models to further explore patient-specific epigenetic programs using ATAC-seq (Fig. 4A). To stratify the OCRs that are lost in P103, we performed K-means clustering including the ATAC-seq data from the three PDOs and the ChIP-seq data from P103 (Fig. 4A). Consequently, we could classify the lost OCRs into (K1) non-enhancer CTCF binding

**Fig. 2 | SMARCB1 reconstitution affects distal enhancers of the MYC oncogene in MRT PDOs. A** Hi-C contact maps depicting the *MYC* locus including the *MYC* promoter and three distal regulatory regions in P103 (E-0.9 Mb; E + 1.1 Mb; E + 1.8 Mb). Left panel depicts chromatin contact in control and the right panel contacts after *SMARCB1* reconstitution. Circles indicate established chromatin loops and dashed circles indicate reduced chromatin loop formation. **B** ATAC-seq, ChIP-seq and CUT&RUN of the indicated proteins in MRT PDO P103 with (SMARCB1 + , lower tracks) and without (Control, upper tracks) *SMARCB1* expression at the *MYC* locus. No antibody was included as negative control for CUT&RUN tracks (BRD9 and SS18). **C** Waterfall plot summarizes changes of the identified chromatin loops before and after *SMARCB1* reconstitution (*n* = 1). The E + 1.1 Mb enhancer (RhOME2) is one of the most affected loops. **D** Z-scores calculated for expression of *MYC*, cell proliferation markers and cell cycle markers in the Control and SMARCB1+ samples of the three PDOs using bulk RNA-seq[11]. **E** CellTiter-Glo assay measuring cell viability of MRT PDOs 7 days after transduction with a non-targeting shRNA (NT) or an shRNA targeting *MYC*, showing that knockdown of *MYC* leads to a decreased proliferation of the tumor PDOs. Data is represented as means ± SD (*n* = 3 independent experiments) (*n* = 3). Source data are provided as a source data file.

sites, (K2) weak enhancers, and (K3) CTCF-independent super-enhancers (Fig. 4A and Supplementary Fig. 5A, B). Consistent with the function of super-enhancers in control of cell identity[39], we identified the SOX protein motifs, including SOX2, SOX9 and SOX17, and functional annotations of many developmental processes that are associated with the K3 cluster-specific open chromatin sites, as well as RhOME2 and 3.1 OCRs (Supplementary Fig. 5C, D). Furthermore, the K3 cluster displayed the highest chromatin occupancy of the ncBAF complex, as measured by BRD9 CUT&RUN, which is dramatically reduced after restoring *SMARCB1* expression (Supplementary Fig. 5A). Whereas the K1 and K2 clusters showed a similar accessibility pattern in all three PDO models, the accessibility loss of the K3 cluster was specific to the P103 line (Fig. 4A). Our Hi-C analysis in P103 showed that these K3 cluster-specific super enhancers form insulation boundaries, which were effectively abolished upon reconstitution of *SMARCB1* expression (Fig. 4B, C). Thus, *SMARCB1* reconstitution in MRT organoids PDOs coincides with dramatic but local, patient-specific changes in genome topology, BAF complex occupancy and enhancer activity.

### MYC enhancer plasticity is reflected in patient MRT

Our data so far raise the intriguing possibility that there is intertumoral heterogeneity on the level of enhancer utilization driving expression of the *MYC* oncogene. To extrapolate our in vitro findings to patient tissues, we first examined publicly available H3K27ac ChIP-seq profiles from patient MRT tissues[40] (Supplementary Data 3). As expected, the *MYC* promoter is marked by high levels of H3K27ac in the majority of patient samples, indicative of active transcription (Fig. 5A). Moreover, broad enrichment for H3K27ac could be detected at RhOME1-3, showing that tumor super-enhancer landscapes identified in MRT PDOs are preserved in MRT tissues. No clear peaks at RhOME1-3 could be detected in STJ0090 and STJ0537, while weak H3K27ac signals were detected at the *MYC* promoter. Remarkably, and in line with our PDO data, the presence of H3K27ac peaks at the different RhOME loci was heterogeneously distributed across the other patient samples (Fig. 5A).

To assess whether differential enhancer utilization occurs within a tumor, i.e., intratumoral heterogeneity, we performed combined single-cell RNA-seq and ATAC-seq on seven MRT patient tissue samples using single-cell Multiome (10x Genomics) (Fig. 5B). After filtering, 14,799 cells were left for further analyses, with a median of 2040 cells per tumor (Fig. 5C). Normal and tumor cells were assigned by cell type marker genes[41] and *SMARCB1* expression (Supplementary Fig. 6A–D). Uniform Manifold Approximation and Projection (UMAP) space reveals that tumor cells are clustered by patient, while non-malignant cells are clustered by cell type (Fig. 5C). *MYC* expression, as well as OCRs at the *MYC* promoter were found across all MRTs and some normal cell clusters (Fig. 5D and Supplementary Fig. 6E). However, despite a detectable *MYC* promoter signal, almost no accessible chromatin could be detected at RhOME1-3 in normal cells (Fig. 5D), suggesting that these super-enhancers are tumor-specific in at least these samples. Crucially, we observed different combinations of OCRs at RhOME1-3 in the different patient MRTs (Fig. 5D, E). More specifically, while P156 and P168 are exclusively defined by OCRs at RhOME2, P041 and P052 primarily show OCRs at RhOME1 and RhOME3, and in P166 OCRs were only detected at RhOME3. No OCRs were found in any of the RhOMEs in P116 and P138, suggesting supraphysiological

activation of *MYC* occurs through regulatory elements other than RhOME1-3 in these tumors. Collectively, these results show the existence of intertumoral heterogeneity on the level of super-enhancer activity driving expression of the *MYC* oncogene in MRT.

### ncBAF inhibition induces gene expression signatures that resemble SMARCB1 reconstitution

It was recently demonstrated that MRT may be driven by the ncBAF complex aberrantly localizing at super-enhancers in MRT cells[40,42,43]. As such, inhibition of the ncBAF subunit BRD9 was demonstrated to be a putative therapeutic vulnerability in MRT. We therefore set out to investigate the effect of ncBAF inhibition on MRT PDOs (Fig. 6A). To do so, we treated MRT PDOs with a pharmacological inhibitor of BRD9 (I-BRD9)[42–44]. We observed that, morphologically, MRT cells exhibited a differentiation phenotype similar to *SMARCB1* reconstitution[11] (Fig. 6A) and cell growth was significantly inhibited (Supplementary Fig. 7A). Quantitative RT-PCR (RT-qPCR) confirmed that BRD9 inhibition caused a significant decrease in *MYC* mRNA levels (Fig. 6B). We performed RNA-seq to further measure the transcriptomic changes following ncBAF inhibition and found a significant association between gene expression changes induced after I-BRD9 treatment and those upon *SMARCB1* reconstitution (Fig. 6C). To statistically confirm the overlap between the differential gene sets, we performed a Fisher's exact test on the two sets (Fig. 6D), confirming strong similarity. Consistent with a decrease in proliferation, we observed a downregulation of cell cycle-related gene sets as well as MYC target genes (Fig. 6E). In the upregulated gene sets we found an enrichment of genes associated with differentiation (Fig. 6E), consistent with the morphological phenotype of the cells treated with the BRD9 inhibitor (Fig. 6A). CUT&RUN for BRD9 and SS18 confirmed decreased binding of the ncBAF complex at the *MYC* promoter as well as RhOME2 and 3 loci (Fig. 6F). More generally, a genome-wide loss of binding of BRD9 and SS18 was observed after I-BRD9 treatment (Supplementary Fig. 7C, D), which was not caused by a treatment-induced decrease in BRD9 and SS18 protein expression (Supplementary Fig. 7B). Therefore, confirming treatment-induced loss of ncBAF complex binding. Thus, inhibition of the ncBAF complex in MRTs, eliminating all three BAF complexes, phenocopies the effects of reconstitution of *SMARCB1* expression that restores all the BAF complexes.

Altogether, our comprehensive study shows that derailed activity of super-enhancers as a consequence of *SMARCB1* loss underpins MRT tumorigenesis and serves as a blueprint for unraveling the contribution of BAF complex mutations to tumorigenesis across cancers.

## Discussion

Decades of cancer genetics research have revealed that adult tumors are unique based on their genetic profile, with each patient tumor having a distinctive mutation landscape[45,46]. Although adult cancers can harbor hundreds to thousands of mutations, recurrent driver mutations can be identified[47]. Typically, specific signaling pathways affecting cell proliferation and survival can be affected by mutations in different pathway members. Pediatric tumors on the other hand have a low mutational burden and are characterized by a small number or even a single driver event[48]. Additional epigenetic changes may

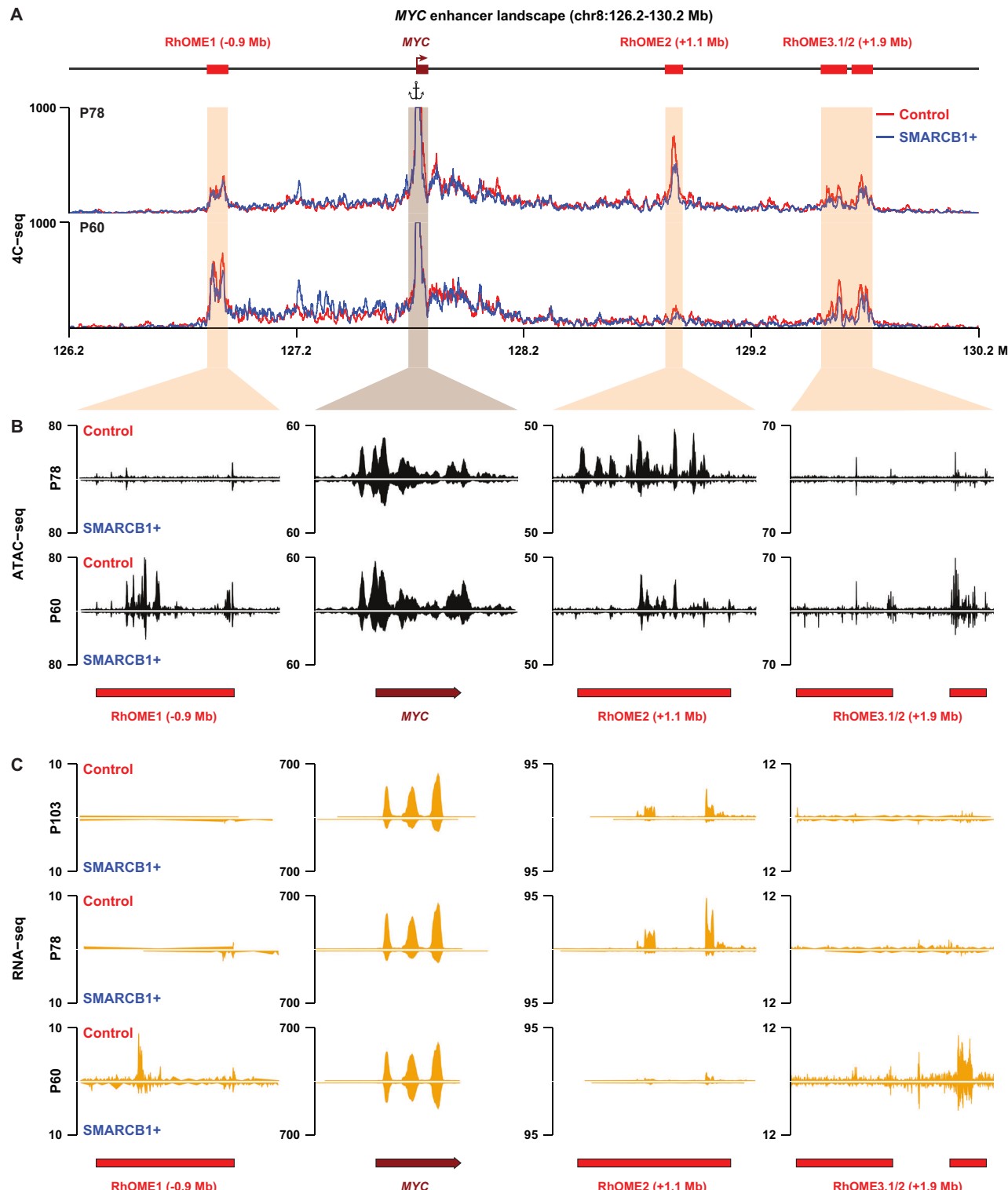

**Fig. 3 | Identification of patients-specific *MYC* enhancer-ome in different MRT PDOs. A** Contact map of 4C-seq data identifies chromatin interactions between the *MYC* promoter and three Rhabdoid Oncogenic *MYC* Enhancers (RhOMEs) in two additional MRT PDO models (P60 and P78; *n* = 2). **B** Chromatin accessibility and (**C**) enhancer RNA (eRNA) expression at the three RhOMEs and the *MYC* mRNA expression in the indicated MRT PDOs with (SMARCB1 + ) and without (Control) *SMARCB1* expression.

contribute to pediatric tumor formation. Identifying recurrent patient-specific epigenetic driver landscapes has remained challenging. Because PDOs maintain many molecular characteristics of their parental tumors they can be used to study patient-specific changes on the epigenetic level as well. Our BAF complex reconstitution experiments enabled us to zoom into putative oncogenic enhancers and the role of

the distinct BAF complex compositions at these sites. Upon analyzing the (single-cell) regulatory landscapes of primary MRT tissue we were able to identify potential super-enhancers that were recurrently activated in a patient-specific manner.

A common denominator of MRT is the high expression of the *MYC* oncogene[11,12] The regulatory mechanisms causing aberrant *MYC*

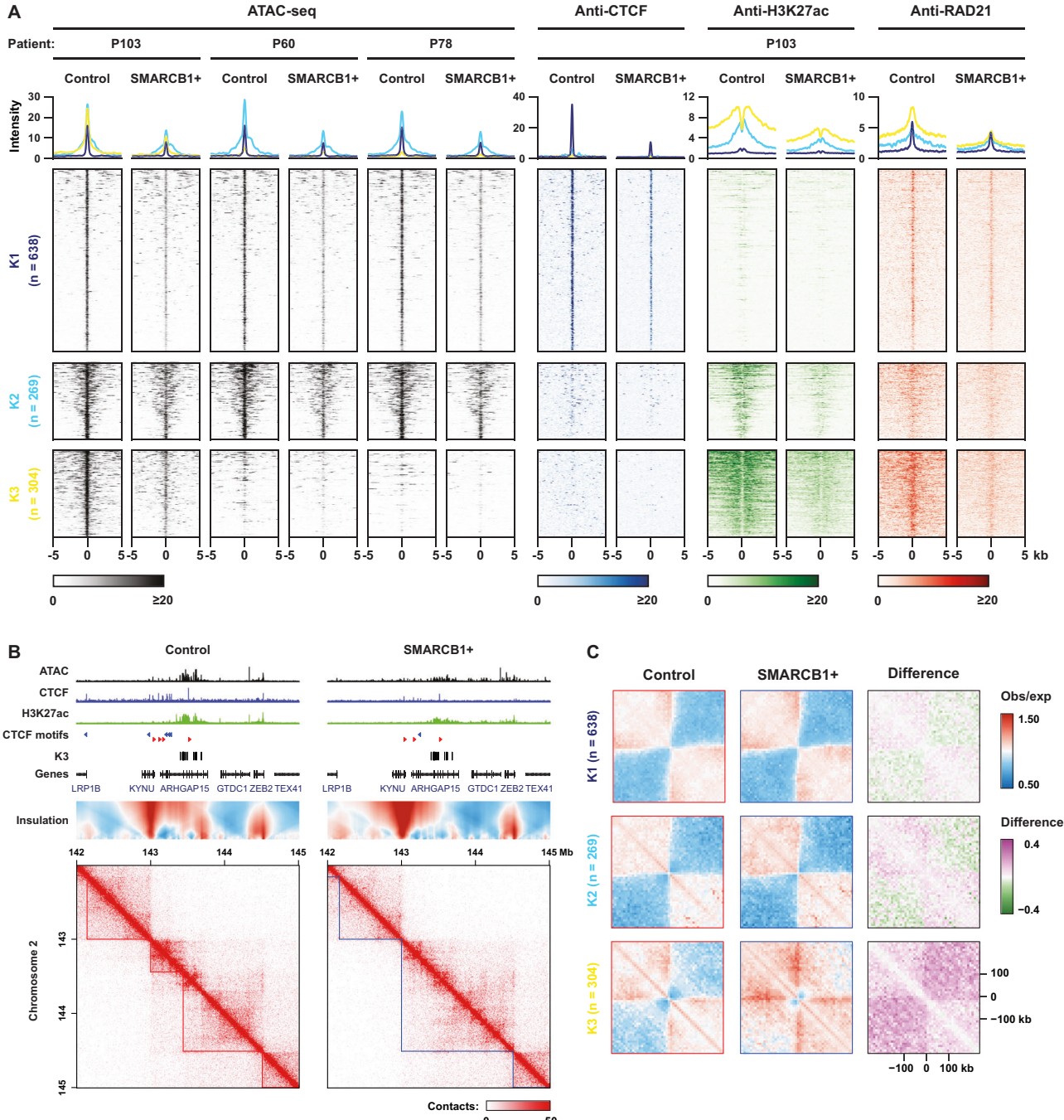

**Fig. 4 | Patient-specific super enhancers form TAD boundaries in the SMARCB1-null tumor PDOs that are independent of CTCF. A** A K-means clustering analysis on the control-specific (lost) OCRs, based on their chromatin features in these PDOs (ATAC *n* = 3, CTCF ChIP *n* = 1, RAD21 ChIP *n* = 1, H3K27Ac ChIP *n* = 1). **B** Contact map of P103 Hi-C data depicting an example of a super-enhancer forming

a TAD boundary (left panel), that is diminished after *SMARCB1* reconstitution (right panel). Supporting tracks of ATAC-seq and ChIP-seq (CTCF, H3K27Ac) are displayed above. **C** Aggregate Region Analysis (ARA) showing the insulation borders in each separate K-mean cluster. The K3 cluster is comprised of chromatin loops with SMARCB1-sensitive insulation borders in P103.

activation in MRT have so far remained unknown as no recurrent genetic alterations besides *SMARCB1* loss have been described[6,12–14]. Here, we identify three putative super-enhancers involved in *MYC* regulation in MRT. *SMARCB1* loss in these tumors leads to increased looping of these enhancers to the *MYC* promoter, potentially activating its transcription. While different RhOMEs loop to the *MYC* promoter in different tumors (intertumoral heterogeneity), we find preferential use of one or a combination of RhOMEs within the same tumor. Further mechanistic studies are required to further elucidate the super-enhancer function in MRT development in more detail. The patient-specific enhancer landscapes found in MRT could reflect the developmental origin of MRT, which lies in the neural crest[11]. Neural crest development is characterized by rapid switching of cell states caused by, amongst others, chromatin reorganization to assure quick and simultaneous development of several different cell types[49–51]. We hypothesize that loss of *SMARCB1* in a specific cellular context during neural crest development prevents the inactivation of certain *MYC* enhancers, which is essential for proper lineage specification. Ultimately, these *MYC* enhancers may transform to an oncogenic *MYC* super-enhancer and drive tumorigenesis. Depending on the cellular

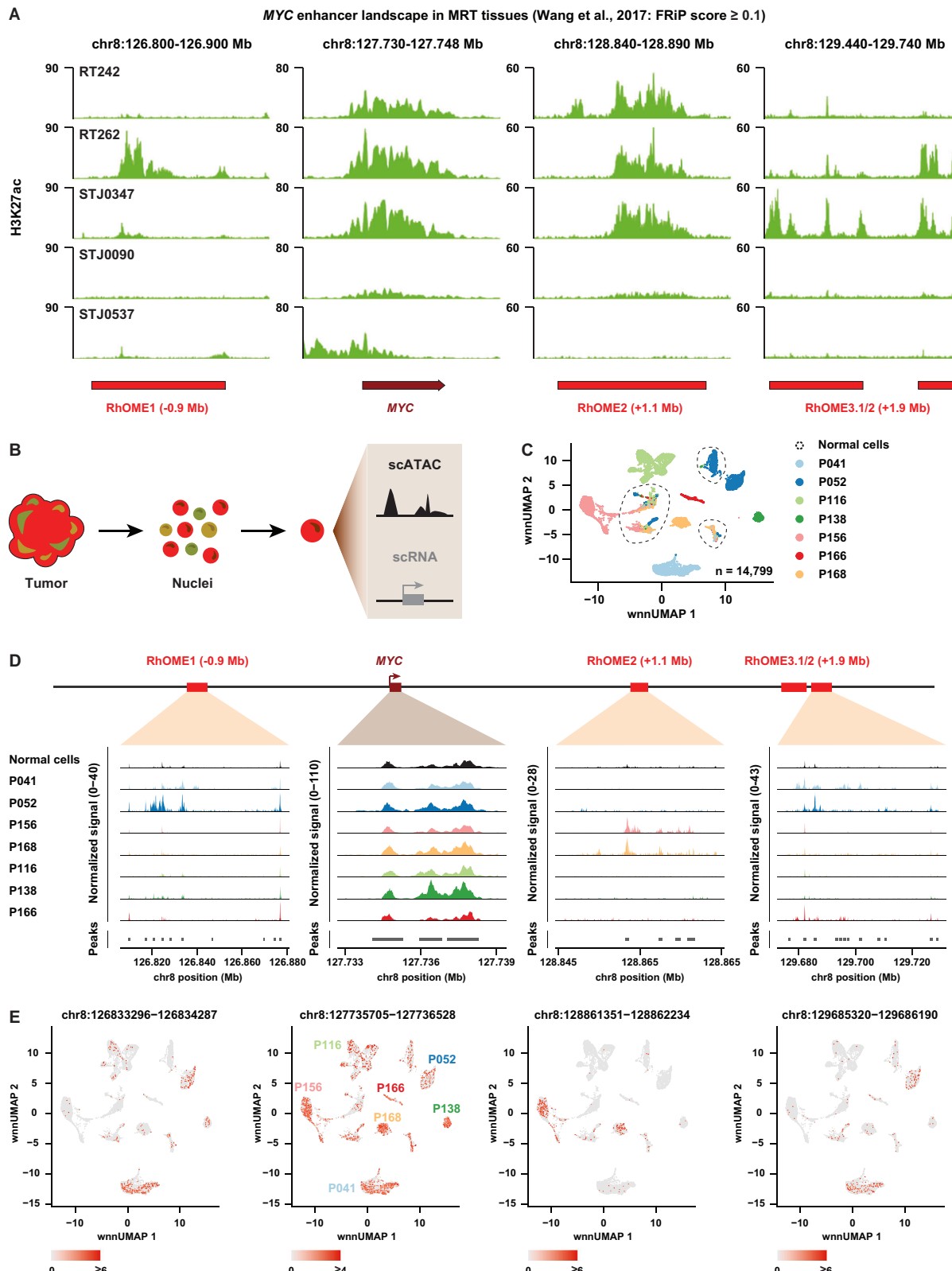

**Fig. 5 | Analyses of MRT tissues reveal inter- and intra-tumor heterogeneity of *MYC* enhancer landscape. A** Genomic tracks depicting H3K27Ac signal at the *MYC* locus from publicly available H3K27Ac ChIP-seq data[40]. Data was filtered based on an average FRiP score of higher than 0.1. **B** Acquisition of single-cell Multiome data (gene expression + ATAC-seq) from MRT tissues using Chromium 10x Genomics platform. **C** UMAP discriminating tumor cells, but not normal cells, from the MRT tissues derived from different patients (*n* = 1 per patient). Tumor cells cluster separately while normal cells are depicted as a mixed population from different patients. **D** A pseudo-bulk view of scATAC-seq reads at the *MYC* locus for 7 MRT patients. **E** UMAP depicting RhOME specific peak signal within each single cell (*n* = 1 per patient).

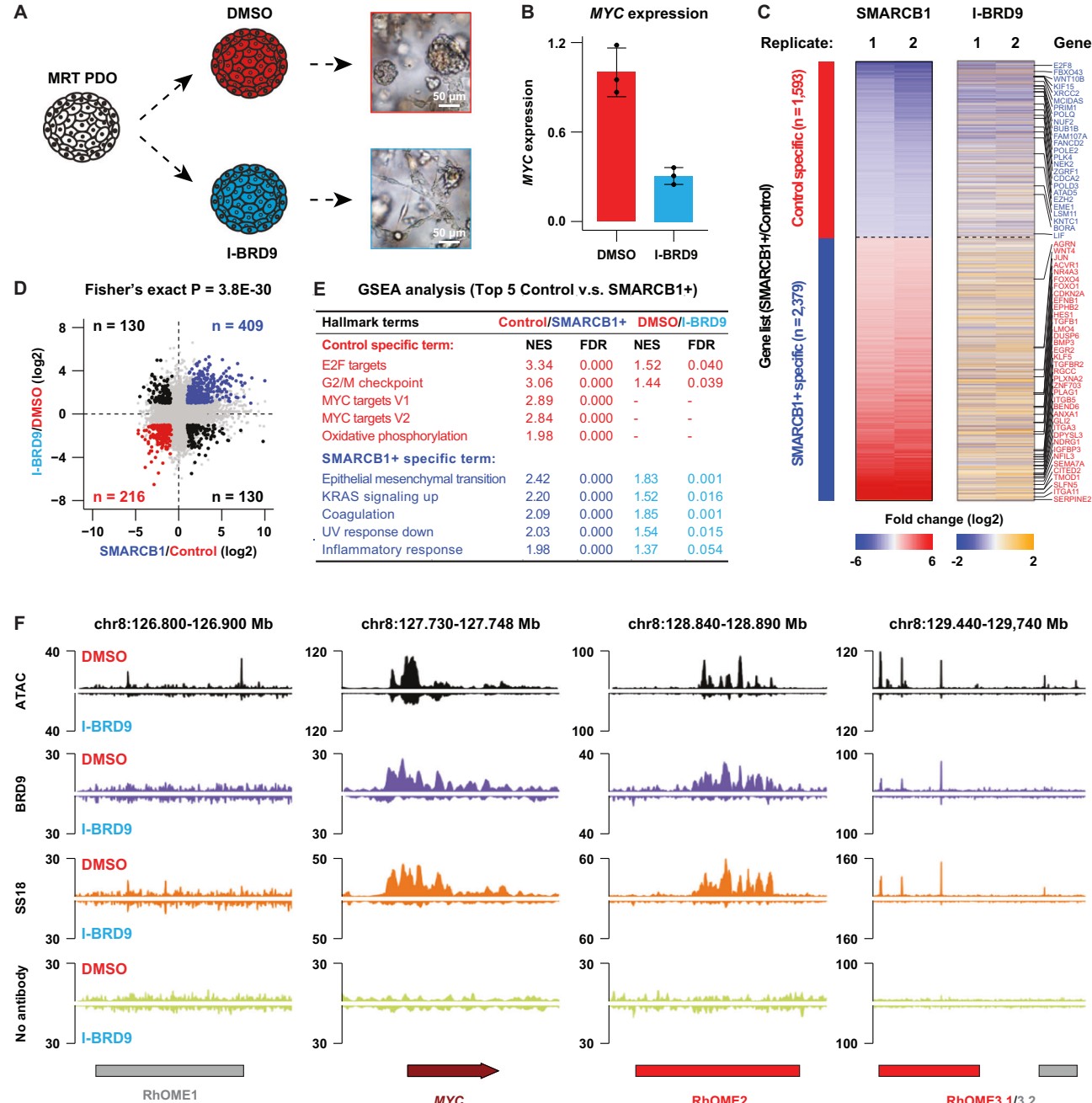

**Fig. 6 | A therapeutic strategy for MRT patients via targeting a ncBAF subunit named BRD9 using a pharmacological inhibitor. A** Schematic representation of the experiment and brightfield images showing a differentiation phenotype upon BRD9 inhibition in MRT PDOs (adapted from Custers et al.[1]). **B** RT-qPCR for *MYC* in P103 treated with DMSO or I-BRD9 (10 µM). Data is represented as means ± SD (*n* = 3 independent experiments). Source data are provided as a source data file. **C** Heatmaps showing the transcriptomic changes of differential expressed genes between SMARCB1+ and Control in either *SMARCB1* reconstitution or BRD9 inhibition experiments (*n* = 2). **D** Correlation of transcriptomic changes of

SMARCB1 reconstitution with BRD9 inhibition. Upper right panel show all genes significantly upregulated in both experiments (blue dots) (*n* = 2). Lower left panel represent all genes specific for control condition in both experiments (red dots). Statistical significance was calculated by a fisher's exact test. **E** GSEA analyses showing top five hallmark gene sets identified in the *SMARCB1* reconstitution and the corresponding differential expression analysis after BRD9 inhibition. **F** Chromatin occupancy at the *MYC* locus 5 days after I-BRD9 treatment depicting ATAC-seq and CUT&RUN for BRD9 and SS18. No antibody was included as the negative control.

identity of the tumor-initiating cell, different enhancer landscapes may be active, possibly explaining why we find patient-specific enhancers. The use of these super-enhancers is identified from a heterogeneous population of cancer cells, suggesting also distinct usage and activity of these super-enhancers within one patient. The degree of heterogeneity in chromatin accessibility within one patient remains to be further investigated. During development, cellular identity is in part determined by growth factors and morphogens that signal to transcription

factors to establish a cell-type-specific regulatory landscape. How this regulatory landscape is maintained in the absence of these signaling molecules merits further investigation. We highlight how *MYC* over-expression can be explained by the accessibility and activity of patient-specific super-enhancers. Although we only exemplify this in MRT, the concept of patient-specific epigenetic regulation of oncogenic drivers may be applicable to a broader range of tumor entities and pave the path toward more specific epigenetic drugs in cancer treatment.

Our work reveals a role for SMARCB1 and the ncBAF complex in 3D genome organization regulating long-range promoter-enhancer loops. Furthermore, SMARCB1 is required to prevent the formation of super enhancer-driven TAD boundaries, which are observed in MRT cells. In a previous study, restoration of SMARCB1 in an epithelioid sarcoma cell line did not show any apparent effects on 3D genome organization[52]. Although in these cells *MYC* interacts with the genomic region of RhOME3, no difference was observed upon *SMARCB1* reconstitution[52]. We hypothesize that SMARCB1 executes its role in 3D genome organization by preventing the formation of ectopic super-enhancers which may be cell-type specific. Furthermore, our data suggest that the loss of SMARCB1 in MRT causes increased binding of the ncBAF complex at super-enhancers. This does not seem to be caused by binding by other BAF complexes, but rather driven by a shifted balance of BAF complex compositions in a SMARCB1 proficient or deficient context (Supplementary Fig. 8). Super enhancers are bound by cohesin and we also observed this in the MRT cells explaining how the very distal RhOMEs can interact with the *MYC* promoter. We and others have previously shown that accessible chromatin regions, which are directly regulated by the BAF complex, are important for cohesin binding at active enhancers[28,53,54]. Therefore, *SMARCB1* loss may alter long-range chromatin interactions via reshaping cohesin occupancy in the mutant cells. *SMARCB1* reconstitution drastically reduced CTCF binding at certain sites such as the *MYC* locus, but the CTCF-anchored chromatin loops and the majority of TAD boundaries remained mostly unaffected. This is in line with previous literature showing that at least 85% of CTCF depletion is needed to induce major changes in 3D genome organization[55]. Consistent with a model of cohesin-mediated loop extrusion we hypothesize that cohesin is loaded at the distal super-enhancers and brings the enhancer and the promoter in proximity. In addition, the reported enhancer docking CTCF site in the *MYC* promoter[56] may play a role in the interactions between the *MYC* promoter and the RhOMEs. One possible explanation for the loss of chromatin loop formation at the *MYC* locus is a decreased binding of CTCF-independent cohesin, which plays a critical role in regulating promoter-enhancer loops[28]. We indeed observed high binding of cohesin at RhOMEs, possibly creating a loading site. To further understand enhancer selection in MRTs, a mechanistic study is required to delineate the regulome for each of the *MYC*-RhOME loops.

By inhibiting BRD9, and therefore repressing the binding and activity of the ncBAF complex, we could show that loss of the ncBAF complex has a similar effect as rescuing the cBAF complex. Therefore, MRT initiation during development might be dependent on the binding at, and activity of the ncBAF complex at chromatin regions regulating oncogene expression thereby driving tumorigenesis. In addition to MRTs, several other pediatric cancers are characterized by aberrations in the BAF complex[5]. Moreover, BAF complex members are among the most commonly mutated genes in adult malignancies[4]. Using our study as a blueprint for other BAF complex-mutant cancers could shed light on the importance of intertumoral epigenetic heterogeneity to tumorigenesis and the therapeutic potential of BAF complex targeting on a broader scale.

## Methods

### Ethics statement

MRT tissues used for single-cell Multiome (Supplementary Data 4) were obtained under approval by the medical ethical committee of the Erasmus Medical Center (Rotterdam, the Netherlands) and the Princess Máxima Center for pediatric oncology (Utrecht, the Netherlands). Written informed consent was provided by all patients and/or parents/guardians. Approval for use of the subject's tissue samples and clinical data within the context of this study has been granted by the Máxima biobank and data access committee (https://research.prinsesmaximacentrum.nl/en/ core-facilities/biobank-clinical-research-committee)   (biobank request nr. PMCLAB2018.005).

### Cell Culture

**Patient-derived organoids.** MRT PDOs were cultured as previously described[23]. For BRD9 inhibition experiments, MRT PDOs were mechanically dissociated and plated at a density of 10.000 cells/μl in 10 μl BME droplets. After 24 h, 10 μM of I-BRD9 (HY-18975; MedChemExpress) was added to the PDOs. PDOs were harvested 120 h later for RNA isolation.

**G401 cell line.** G401 cells were cultured in DMEM, high glucose (11965084, Thermo Fisher) supplemented with 10% FBS and 1% Penicillin-Streptomycin, passaged 1:15 twice a week and frequently tested for mycoplasma. Lentiviral transductions were performed identical to patient-derived organoid cultures.

### Vector construction

pLKO.1-UbC-luciferase-blast and pLKO.1-UbC-hSMARCB1-blast lentiviruses were previously described[11]. For *MYC* knockdown experiments, the following targeting sequence was used: 5'-CCCAAGGTAGT-TATCCTTAAA-3'.

### Lentiviral transductions

Lentiviral transductions were performed as described[11,57]. For *SMARCB1* reconstitution, MRT PDOs were transduced with pLKO.1-UbC-luciferase-blast or pLKO.1-UbC-hSMARCB1-blast lentiviruses, as described[11]. PDOs were dissociated into single cells and transduced via spinoculation at 32 °C for one hour (600 x g) with a virus MOI of ± 0.4. After four hours of recovery at 37 °C, cells were re-seeded in basement membrane extract (BME). After two days, 10 μg/ml blasticidin was added to the culture medium. After selection (i.e., four days after transduction when all non-transduced control cells died), cells were harvested for the different applications. For *MYC* knockdown experiments, MRT PDOs were transduced with pLKO.1-puro lentiviruses. For each condition, 30,000 cells were seeded. After two days, puromycin was added to the culture medium. Nine days after lentiviral infection, PDOs were harvested for RNA extraction.

### Quantitative RT-PCR

PDOs were harvested in RA1 buffer and RNA was subsequently isolated using the Macherey-Nagel RNA isolation kit following the manufacturer's instructions. The extracted RNA was used for cDNA production using GoScript Reverse Transcriptase (Promega). Quantitative RT-PCR was performed using IQ SYBR green mix (Biorad) following manufacturer's instructions. Results were calculated using the ΔΔCt method. Primer sequences: *MYC*_Fw: 5'-GATTCTCTGCTCTCCTCGACG-3', *MYC*_Rv: 5'-GATGTGTGGAGACGTGGCA-3', *GAPDH*_Fw 5'-TGCAC-CACCAACTGCTTAGC-3', *GAPDH*_Rv 5'-GGCATGGACTGTGGTCAT-GAG-3'.

### Cell viability assay

CellTiter-Glo (CTG) assay was performed to assess cell viability based on the quantitation of ATP. Cell growth was assessed either 7 days after *MYC* knockdown or 5 days after starting I-BRD9 or BI-9564 treatment in MRT PDOs or G401 cells. Cell-Titer-Glo 3D reagent (Promega) was performed according to manufacturer's instructions.

### Western Blot

Cells were harvested in KLB Lysis buffer and protein was quantified using Pierce™ BCA Protein Assay Kit. Blocking of nitrocellulose membrane was performed in either 5% BSA or 4% ELK, 1% BSA for 1 hour. The following primary antibodies were used: anti-cMYC (Cell Signaling, 13987 S), anti-BRD9 (Proteintech, 24785-I-AP), anti-SS18 (Cell Signaling, 21792 S) and anti-GAPDH (Abcam, ab9485).

## RNA-sequencing

RNA was isolated from the harvested PDOs using TRIzol RNA isolation reagent (Invitrogen) according to the manufacturer's instructions. RNA-seq libraries were prepared using the TruSeq Stranded RNA LT Kit (Illumina). The libraries were sequenced on an Illumina NextSeq 550 platform using the paired-end 44 + 32 (75 cycles) mode. Two biological replicates were generated for each of the RNA-seq experiments in this study.

RNA-seq data were mapped against the hg38 reference genome by TopHat2 (version 2.1.1)[58]. Mapped reads with a mapping quality score <10 were removed using SAMtools. The read coverage of exons for each gene in the Homo Sapiens GRCm38.92 annotation file was determined with the HTSeq tool (version 0.9.1)[59]. The coverage files were generated with the 'normalize to 1× genome coverage' methods in deepTools. For visualization, we combined the two biological replicates of each condition to generate one merged bigWig file. The filtered coverage data was normalized using DESeq2 (version 1.24.0) with default parameters[60]. Differential peaks were detected using a Wald test (FDR < 0.05 and fold change ≥2). Gene set enrichment analysis was performed with a desktop version of the GSEA tool (version 4.1.0)[61] and the Molecular Signatures Database (MSigDB; v2022.1)[62].

## ATAC-sequencing

PDOs were washed in ice-cold AdDF + ++ and viably frozen in Recovery Cell Culture Freezing Medium (Thermo Fisher). For library preparation, cells were thawed and processed following an established protocol. In short, nuclei were isolated from cells and permeabilized. The isolated nuclei were tagmented using Tn5 transposase (Illumina, Nextera DNA Library Preparation Kit), followed by two sequential 9-cycle PCR amplification steps. The resulting DNA fragments (<700 bp) were purified using SPRI beads (Beckman). ATAC-seq libraries were sequenced on a HiSeq 2500 (Illumina). For each of the ATAC-seq experiments, the data were recorded in biological triplicates.

ATAC-seq data were analyzed as previously described. In short, sequencing reads were mapped to the hg38 reference genome using BWA-MEM (version 0.7.15-r1140)[63]. The mapped reads were filtered using SAMtools[64], discarding reads with mapping quality score <15, as well as optical PCR duplicates. The coverage files were produced using the deepTools (version 3.0) method "normalize to 1X genome coverage"[65]. For visualization, we combined the biological triplicates of each condition to generate one merged bigWig file. A merged peak list was generated from ATAC-seq data of MRT control and *SMARCB1*+ cells (*n* = 3 independent experiments). The read coverage under the peaks was determined using a HTSeq tool (version 0.9.1)[59]. The peaks with ≥10 in each replicate were included for further analysis. The filtered coverage data was normalized using DESeq2 (version 1.24.0) with default parameters[60]. Differential peaks were detected using a Wald test (FDR < 0.05 and fold change ≥2). Functional annotation of the differential peaks was performed using the GREAT analysis tool (version 4.0.4)[66].

## (Double crosslinked) ChIP-sequencing

The ChIP-seq experiments were performed according to an established protocol[28]. PDOs were washed in ice-cold AdDF + ++ and PBS. To perform a double-crosslinking reaction, a first cross-linking was performed by adding 2 mM disuccinimidyl glutarate (DSG, 20593, Thermo Fisher) for 45 minutes at room temperature and washed once with 1x PBS before second cross-linking reaction. Subsequently, cells were cross-linked with a final concentration of 1% formaldehyde for 10 minutes. Glycine (2.0 M) was used to quench the cross-linking reaction. The cross-linked cells were then lysed and sonicated using Bioruptor Plus sonication device (Diagenode) to obtain ~300 bp chromatin. For ChIP, sonicated chromatin was incubated overnight at 4 °C with antibodies that had first been coupled to Protein G beads

(Thermo Fisher). After incubation, captured chromatin was washed, eluted and de-crosslinked. The released DNA fragments were purified using MiniElute PCR Purification Kit (Qiagen). The ChIP experiments were performed using the following antibodies: CTCF (07-729, Merck Millipore, 5 μl per ChIP), RAD21 (ab154769, Abcam, 2.2 μl per ChIP), SMARCB1 (91735 S, Cell Signaling, 5 μl per double crosslinking ChIP) and H3K27ac (ab4729, Abcam, 5 μl per ChIP). The purified DNA fragments were prepared using the KAPA HTP Library Preparation Kit (Roche) or MicroPlex Library Preparation Kit v3 (only SMARCB1 double crosslinked ChIP, Diagenode) following manufacturer's instructions. The libraries were sequenced on an Illumina HiSeq 2500 using the single-end 65-cycle mode.

ChIP-seq data were analyzed as previously described. In short, sequencing reads were mapped to the hg38 reference genome using the Bowtie 2 mapper (version 2.3.4.1)[67]. The mapped reads were filtered using SAMtools, discarding reads with mapping quality score <15, as well as optical PCR duplicates. The coverage files were produced using the deepTools (version 3.0) method "normalize to 1X genome coverage"[65].

## CUT&RUN

CUT&RUN experiments were performed as previously described[68]. Primary antibodies were incubated overnight with the following conditions (per 500.000 cells): no-antibody control, anti-BRD9 (24785-I-AP, Proteintech, 1:400), anti-SS18 (21792 S, Cell Signaling, 1:160). FACS sorting was performed on nuclei, retrieving a DAPI (D3571, Thermo Fisher) positive and GhostDye negative population (#18452 S, Cell Signaling). For *SMARCB1* reconstitution samples, a total of 1600 cells were sorted per condition, whereas for I-BRD9 treatment 5500 cells were sorted and used for downstream library preparation. Sequencing was performed on an Illumina NextSeq2000 using paired-end 100 cycles (2x50bp).

CUT&RUN data were analyzed using the same pipeline as ATAC-seq. In short, sequencing reads were mapped to the hg38 reference genome using BWA-MEM (version 0.7.15-r1140)[63]. The mapped reads were filtered using SAMtools[64], discarding reads with mapping quality score <15, as well as optical PCR duplicates. The coverage files were produced using the deepTools (version 3.0) method "normalize to 1X genome coverage"[65].

## Hi-C and 4C-seq

We generated Hi-C data as previously described with minor modifications[28]. In short, MRT PDOs (~10 million cells) were washed in ice-cold AdDF + ++ and PBS. Subsequently, cells were cross-linked with a final concentration of 2% formaldehyde for 10 min. Glycine (2.0 M) was used to quench the cross-linking reaction. The restriction enzyme MboI was used to digest cross-linked DNA in the nucleus. At the restriction overhangs, biotinylated nucleotides were incorporated. Subsequently, overhangs were joined by blunt-end ligation. The ligated DNA was enriched by streptavidin pull-down. A standard end-repair and A-tailing method was used to further prepare the Hi-C libraries, which were sequenced on an Illumina Nova-Seq platform generating paired-end 150 bp reads.

Hi-C sequencing reads were processed using HiC-Pro[69],which includes mapping, identification of valid Hi-C pairs, generation of contact matrices and ICE normalization. HiCCUPS (version 0.9) was used to call chromatin loops. Subsequent analyses were performed in GENOVA, a visualization tool of Hi-C data written in R[70].

4C was performed as previously described[28]. In short, we used MboI as the first and Csp6I as the second restriction enzyme. The viewpoint was designed at the *MYC* promoter region using the primer pair: "CTCTTTCCCTACACGACGCTCTTCCGATCTTCTCCCTGGGACTC TTGATC" and "ACTGGAGTTCAGACGTGTGCTCTTCCGATCTGTCTGT TTAGCCCTGAGATG". The 4C-seq libraries were sequenced using a

NextSeq 550. Two biological replicates were measured for each of the 4C experiments in this study.

Mapping of the sequencing reads was performed according to our 4C mapping pipeline (http://github.com/deWitLab/4C_mapping). 4C data was normalized to 1 million intrachromosomal reads using peakC[71]. For visualization, we combined the two biological replicates.

## Motif analysis
Motif enrichment was computed using a similar method described in our previous publication[28]. We first identified and quantified the number of motifs for the peaks specific to MRT control or *SMARCB1+* samples using GimmeMotifs (version 0.13.1) and the non-redundant cis-bp database (version 3.0)[72]. As a background peak set, we used the peaks that were unchanged upon *SMARCB1* re-expression. First, we normalized motif frequencies to the total number of identified motifs in that sample. Then, we calculated the log2-enrichment score of MRT control or *SMARCB1+* motifs by comparison to motif frequency in the background peak set. The p-value was calculated using the Fisher's exact test.

## Single-cell Multiome ATAC + GEX
MRT tissues were processed using standard tissue processing procedure following the 10x Genomics protocol (CG000338 Rev D). In brief, tissues were minced into small pieces and homogenized using a dounce tissue grinder. After cell lysis, the sample was filtered once using a 70 μM filter as well as a 40 μM filter. Intact single nuclei were sorted (Supplementary Fig. 6F) and two independent samples were mixed based on different gender (male with female). Between 2.000 to 40.000 nuclei were loaded on the Chip J Chromatin Controller (10x Genomics). Library preparation of gene expression (GEX) as well as ATAC library was performed following manufacturer instructions (10x Genomics). Libraries were sequenced on the Illumina platform Nova-Seq6000 paired-end according to manufacturer specifications (10x Genomics). Initial processing of raw data files was done by cell ranger-arc (version 2.0.0), seurat (version 4.1.1)[73] and signac (version 1.7.0)[74].

Reads were mapped to GRCh38 and cell genotypes were annotated by souporcell[75]. Cell ranger-arc aggr function was used to harmonize detection of peak calling. Further filtering steps were included to filter cells of good quality (ATAC counts <50000 & > 800, RNA counts <30000 & > 800, nucleosome signal <1.5, TSS enrichment > 1, % of blacklist regions <3%, % of mitochondrial genes <20%). Data was normalized using SCT normalization (sctransform version 0.3.5)[76] for RNA-seq data and term frequency-inverse document frequency (TF-IDF) normalization for ATAC-seq data (signac version 1.7.0)[74]. Gene marker expression (SingleR version 1.10)[77] was used to annotate normal cell genotypes compared to tumor cells. Human Primary Cell Atlas Data was used as cell type reference (celldex version 1.6.0)[41,77] for normal cell annotation. Normal versus tumor cell identification was verified by the absence of *SMARCB1* gene expression. UMAP plots were generated via joined clustering of both datasets (PCA 1:25, LSI 2:30).

## Reporting summary
Further information on research design is available in the Nature Portfolio Reporting Summary linked to this article.

## Data availability
The raw sequencing data generated in this study (RNA-seq, ATAC-seq, ChIP-seq, CUT&RUN, 4 C and HiC data) have been deposited in the European Genome-phenome Archive (EGA. www.ebi.ac.uk/ega/) under accession code EGAS00001007590. To protect patient privacy, as required by law, access to the sequencing data deposited in the EGA is controlled by the Data Access Committee (DAC) of the Princess Maxima Center. All researchers can obtain access by submitting a project proposal to the DAC (biobank@prinsesmaximacentrum.nl). Requests will be handled within ~2 weeks. The DAC will also determine the length of permitted access. The processed sequencing data of RNA-seq, ATAC-seq, ChIP-seq and HiC are deposited at GEO (www.ncbi.nlm.nih.gov/geo/) under accession code GSE218115. The processed sequencing data of combined single-cell RNA and ATAC data have been deposited at GEO under accession code GSE218385. The H3K27ac ChIP-seq publicly available data used in this study are available under accession code GSE71506 (https://www.ncbi.nlm.nih.gov/geo/query/acc.cgi)[40]. Publicly available bulk RNA-seq data from kidney tumors is available under accession code EGAD00001005318[23]. Source data are provided with this paper. The remaining data are available within the article, supplementary information, and source data. Source data are provided with this paper.

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

## Acknowledgements

We acknowledge funding from the European Research Council (ERC) (ERC-StG #850571), the Foundation Children Cancer Free (KiKa 338, L.C. and I.P.), and the Dutch Research Council (NWO-Vidi 203.003). Work in the De Wit laboratory was supported by the European Research Council (ERC) (ERC-CoG #865459) and the Dutch Research Council (NWO-Vidi, '016.16.316'). N.Q.L. is supported by a Veni grant from the Netherlands Scientific Organization (NWO, 016.Veni.181.014). Additional support was provided by the American Association for Cancer Research (AACR), the St. Baldrick's Foundation (Pediatric Cancer Research Award to J.D.), Foundation Nikai 4 Life, SNF (P2BSP3-174991, P.Z.), HFSP (LT000209/2018-L, P.Z.), Marie Skłodowska-Curie Actions (798573, P.Z.), and Oncode Institute, which is partly financed by the Dutch Cancer Society. We would like to thank Dr. Gerald Brien for input on the CUT&RUN experiments. We thank the NKI Genomics Core Facility, Oncode's Single-cell (epi) genome sequencing facility (Hubrecht Institute), and the Princess Máxima Center Single Cell Genomics Facility for technical support, as well as the Princess Máxima Center Biobank and Data Access Committee for providing MRT tissues. We are profoundly grateful to the patients and parents who agreed to participate in our research.

## Author contributions

N.Q.L., I.P. and L.C. contributed equally to this work. N.Q.L., I.P., L.C, H.T, P.Z., A.v.O. were involved in data curation and sequencing preparation. N.Q.L. analyzed the data with contributions from I.P., L.C., E.d.W. J.D. I.P., J.H., J.L.B., E.W.H. were involved in tumor sample collection and processing. I.P. analyzed the single-cell data with contributions from D.A and J.D. N.Q.L., I.P., E.d.W. and J.D wrote the manuscript. E.d.W. and J.D. supervised the work. All authors read and approved the manuscript.

## Competing interests

The authors declare no competing interests.
