## [Peer Review File · Nature Communications]

SMARCB1 loss activates patient-specific distal oncogenic enhancers in malignant rhabdoid tumorsReviewers' Comments:

Reviewer #1:

Remarks to the Author:

Malignant rhabdoid tumors (MRT) are aggressive pediatric malignancies characterized by bi-allelic inactivation of SMARCB1. It is remarkable that the genome of MRTs is deficient in other putative known oncogenic drivers and deletions of tumor suppressors, thus suggesting that SMARCB1 loss is the driving mutation. Despite this knowledge and apparent genomic simplicity, the survival of children with MRT is dismal. Herein, the authors use patient-derived organoids (PDO) and multi-omics approaches to define SMARCB1 loss-dependent epigenetic alterations that could provide insight into MRT tumorigenesis. Upon re-introduction of SMARCB1 into MRT PDOs, the authors found prominent looping changes surrounding the MYC promoter driving high MYC expression levels in control samples. Knockdown of MYC revealed reduced MRT PDO proliferation suggesting these alterations are essential to drive MRT cell proliferation. Analysis of additional PDOs revealed patient-specific SMARCB1 loss-dependent looping alterations connecting super enhancers with the MYC promoter. These interactions are termed Rhabdoid Oncogenic MYC Enhancers (RhOMEs) by the authors. Two of the RhOMEs were previously implicated in enhanced MYC levels in tumors while a third is newly identified. Additionally, the heterogeneous enhancer-promoter interactions were found in independent primary patient tumor samples. The authors hypothesized these aberrant topologies could be maintained by residual ncBAF and showed that inhibition of the BRD9 subunit phenocopied SMARCB1 reconstitution, hinting at a mechanism of ncBAF mediated tumorigenesis of MRT. In total, this study provides new insights into the epigenetics of MRT implicating epigenetic regulation of MYC in SMARCB1-deficient tumors.

Specific Comments

1. There is a noticeable lack of hypothesis in the introduction. Hypotheses and support are provided throughout the results but would serve the paper well to be included in the introduction. Also in introduction lines 49-51, the authors make a claim that loss of SMARCB1 is not sufficient for MRT development. This statement is based upon SMARCB1 loss being detected in adjacent normally appearing Schwann cells in MRTs. This observation more likely reflects the lack of transformability of Schwann cells rather than the sufficiency of SMARCB1 loss driving tumorigenesis in the still to be identified cell of origin. In fact, data from a genetically engineered mouse model suggests that *Smarb1* loss is sufficient for MRT development (PMID: 12450796) and should be cited.
2. The state in extended 3B there are 131 loci identified. The authors should include the list of loci and potential interacting promoters. There was a lack of rationale why MYC was chosen other than its MYC. There were no comments about other interactions found besides the those linking super enhancers with the MYC promoter. A general comment about the other interactions would aid in the rigor behind exploring the MYC enhancer-promoter interactions.
3. The authors perform experiments attempting to delete ncBAF activity with BRD9 inhibitors in MRT. These are perplexing given that MRT has been illustrated to be dependent upon ncBAF activity. This work is in cited reference 33. The results should be presented with the context that BRD9 inhibition results in cell death in MRT.
4. Line 212-213, the authors state that they visually inspected the data. This statement provides no benefit to scientific rigor, and actually detracts from the manuscript. The authors should consider altering this language.
5. The data presented in this manuscript does not clearly depict the mechanism in the bottom panel of the cartoon in Figure 7. It is provocative but should be presented as such.

Minor Comments

1. The authors did not comment on the cell type heterogeneity in PDO samples. They performed bulk analyses on the PDO to find tumor cell specific looping interactions, but it remains unclear if PDO are a homogenous population of tumor cells. The authors single cell ATAC analysis of primary patient samples shows that the interactions likely would have been pulled out from a heterogeneous mixture,

but the authors should consider addressing pitfall.
2. References on line 84-85 need formatting.

Reviewer #2:

Remarks to the Author:

In the manuscript „SMARCB1 loss creates patient-specific MYC topologies that drive malignant rhabdoid tumor growth“ Lui et al. characterize SMARCB1-dependent regulatory landscape by using patient derived organoid models. Eventhough the manuscript gives some novel insights into mechanisms of this rare, malignant entity, I have some major concerns, which needs to accepted before acceptance:

Major concerns

- How was lentiviral transduction done? In a PDO consisting of multiple cells? If yes, how was the transduction efficacy? Do we deal with a heterogeneous model consisting of SMARCB1 positive and SMARCB1 negative cells at the same time?
- Why do the authors report first about the findings in one/two PDO and later of some related findings in the other PDOs? This is confusing, especially the finding and conclusion found in these four PDOs are heterogeneous. I would recommend to reconstruct the way of story telling by integrating all PDOs from the beginning.
- As stated before the authors report about heterogeneous findings in four PDOs. Eventhough they validate some findings in patient samples, the questions remains, if it is not possible to increase the number of PDOs for these analyses or at least for parts of these analyzes.
- The author should indicate the number of samples used in this study (e. g. publically available ATAC seq samples, which were reanalyzed).
- I recommend to validate findings of the regulator landscape detected in MRT PDO in patients samples also on other levels, e. g. by including ChIP-seq of CTCF and Hi-C.
- Seven MRT patient samples were used for scRNA-seq, as well as scATAC seq. If I understand correct also in patient sample intertumoral heterogeneity for OCR at RhOME1-3 can be identified. How is this heterogeneity within one tumor (intratumoral heterogeneity)?
- Can the findings of heterogeneous regulator landscape be mechanistically explained? Or are there correlations to genetics (type of SMARCB1 alteration) or clinical differences (e. g. localisation of tumor occurrence)?

Reviewer #3:

Remarks to the Author:

In Liu et al. entitled: “SMARCB1 loss creates patient-specific MYC topologies that drive malignant rhabdoid tumor growth“, the authors utilized malignant rhabdoid tumor (MRT) patient-derived organoid (PDO) models to study how biallelic loss of SMARCB1 may impact gene expression, epigenome, and 3D chromatin topology in MRT. Indeed, SMARCB1 is deleted in >95% of MRT, and it is important to understand how its loss causes cancer. The authors conducted re-expression of SMARCB1 in PDO model (generating a line they called SMARCB1+) and compared it to MRT PDO cells without this protein (WT). They utilized a series of epigenomics assays, including ATAC-Seq, ChIP-Seq, RNA-Seq, Hi-C, 4C, and found that SMARCB1 re-expression globally alters the enhancer landscape (more gain than loss); by Hi-C, they found that while TADs were largely kept, there was chromatin topology difference in the MYC gene loci. The authors further showed that MYC may be activated by different enhancers in a patient-specific manner in different MRT patients; and tried to explore this idea with single-cell multi-omics data from patient’s tumors. Finally, they attempted a therapeutic avenue for MRT by a pharmacological inhibitor of BRD9.

Overall, this paper was well-written with a clear description of the background and methods. Genomics

experiments and analyses were generally performed well. However, I found this paper does not have a central point to make; for several potential points the paper seems to make, the novelty is not strong (see below); it is unclear if any epigenome or chromatin topology change observed is due to direct or secondary effects of SMARCB1 loss/gain; the mechanisms of such changes are also unclear; the main message in the title about patient specific MYC chromatin topology is not fully supported by the data and appeared pre-mature; the treatment by iBRD9 while interesting is not particularly new (and whether the drug is specific to BRD9 or how does it work are unclear, i.e., degrading BRD9 or preventing its chromatin binding).

Main concerns:

1. Novelty issue:

- a. The re-expression of SMARCB1 into its null tumor cells to examine epigenomic changes has been conducted by a few studies (Wang et al., 2017, PMID: 27941797; Nakayama et al., 2017, PMID: 28945250). The novelty of this paper is partially compromised. One potential remaining novelty is that this paper used Organoid system to achieve this goal.
- b. The epigenomic changes of histone marks have been examined previously after SMARCB1 re-expression, even Hi-C was also conducted by Nakayama et al., 2017, PMID: 28945250, although the authors suggested limited changes therein.
- c. The role of BRD9 in helping BAF aberrant chromatin location upon SMARCB1 loss in MRT was reported elsewhere (Wang 2019, PMID: 31015438). Wang et al., not only identified BRD9 as required for MRT growth, but also showed that bromodomain inhibitors of BRD9 did not affect cell growth, implying that BRD9 degradation rather than domain inhibition is required for cell inhibition in MRT. This current paper merely treated PDO with BRD9 inhibitor (a different iBRD9, which may have off targeting to BRD7, see MedChemExpress, HY-18975 information).
- d. This paper conducted Hi-C and found chromatin changes in MYC loci (which may be considered a potential novelty, although global analysis needs some clarification, see below). Unfortunately, the authors provided no explanation of mechanisms of this finding.
- e. This paper used an eye-catching title of patient-specific MYC chromatin topology, but this was only directly showed by Hi-C or 4C in two patient samples (the single cell data will not directly support chromatin topology, at least not at the 3D genome level).

Overall, this paper put in a lot of points that are seemingly novel, but careful inspection of each of these indicates that the novelty is moderate (mostly come from the PDO system, and Hi-C helps somewhat), and some datasets, (e.g., patient 4C/Hi-C and single cell multi-omics data (as resource).

2. Lack of mechanism: it is unclear if any epigenome or 3D chromatin topology change observed in PDO after SMARCB1 re-expression is due to direct or secondary effects of SMARCB1.

- a. The authors need to show where SMARCB1 or other SWI/SNF complex bind in the genome after SMARCB1 re-expression (this is commonly done by many previous papers, Wang et al., 2017, Nakayama et al., 2017). They need to correlate the possible gained/lost binding of SWI/SNF with the altered epigenome in Fig. 1/2/3. At this stage, these are just phenomenon, it is completely unclear how SMARCB1 plays the role directly.
- b. Many of these changes are likely indirect effects/secondary effects after long term SMARCB1 re-expression. A possible strategy is to re-express SMARCB1 with a FKBP12 tag so it can quickly induce or reduce by dTAG to achieve rapid regulation to understand direct roles.
- c. The reasons for changes of the MYC loci Hi-C are also unclear. Did SWI/SNF bind any of these enhancers/promoters directly after re-introduction and then directly suppress MYC? Or did SWI/SNF now gains their "correct" location after SMARCB1 re-introduction, and thus MYC loses SWI/SNF binding at enhancers.
- d. How did the 3D genome change? Did CTCF get directly suppressed by SMARCB1 reexpression at MYC loci?

3. About MYC alteration: Weissmiller et al., 2019 (PMID: 31043611) showed that SMARCB1 re-expression in 293T cells did not obviously impact MYC expression level (Western), but can antagonize MYC binding to chromatin. But this current paper showed that SMARCB1 re-expression strongly

reduced MYC mRNA expression (Fig.3C) in PDO. Is this phenomenon a PDO specific one? Can the authors examine other papers that did SMARCB1 re-expression and see how MYC mRNA and/or protein changes?

4. Ext Data Fig.2A, can the authors show the P(s) curve as commonly used control/treatment two P(s) curves in addition to the log2 fold change curve? The extremely long range curves (40-100Mb) seems to increase a lot. But due to their low frequency in P(s) curve, this may be very little, so P(s) curve is helpful to see.

Ext Data Fig.2D, what is the blue difference/subtraction plot show? Fold change or subtraction?

5. Patient specific MYC topology.

a. The authors conducted Hi-C in one PDO, and 4C-Seq in 2 PDOs. This is too limited data to support the title: patient-specific MYC topologies. Also do the authors mean chromatin 3D topology in the title?

b. Also about the title, the data did not support that "SMARCB1 loss creates patient-specific...". These topologies are more likely due to patient variation, rather than "SMARCB1 loss creates them". Fig.3A shows this well as even with SMARCB1 re-expression the topology still appears different.

c. There is very strong variation of chromatin opening between patients' PDOs – e.g., line 158, "81.6%, 72.1% and 78.7% of the OCRs lost in P103, P78 and P60, respectively, are lost specifically in that PDO". Then it becomes confusing what is the purpose of the K means clustering of ATAC-Seq from 3 PDOs of the changes of P103 (Fig.4)? and to compare to Hi-C seen in P103?

6. The authors then moved to use single cell multi-omics in MRT to make a point that different MRT patients cells may bear different activity of the 3 enhancers close to MYC. Is this a surprise? For example, Corces M et al., 2018, PMID: 30361341 has published ATAC-Seq in hundreds of tumors. If the authors simply select some cancer samples with MYC high expression, is it uncommon that each patient use different ATAC-Seq enhancer activities? Even if so, why is it important biologically or clinically?

7. BRD9 inhibition: the effects on gene expression and cell differentiation are very good. However while Wang et al., (PMID: 31015438) identified BRD9 as required for MRT growth, they also showed that bromodomain inhibitors of BRD9 did not seem to affect cell growth, implying that BRD9 degradation rather than bromodomain inhibition is required for cell inhibition in MRT. This current paper used i-BRD9, what is the rationale for this? Did the authors examine the drugs used by Wang et al. 2019? Did iBRD9 reduce BRD9 protein level, how about BRD7 or BRD4 levels? Is iBRD9 specific to BRD9 or also other bromodomain proteins? These need to be checked to conclude if the effect is solely achieved via BRD9 inhibition.

For BRD9 inhibitor, if the authors want to conclude that it works by preventing residual non-canonical ncBAF complex in MRT tumor cells (with SMARCB1 loss); they need to show ChIP-Seq of SWI/SNF with/without iBRD9, and how they correlate with MYC gene inhibition and the inhibition of MYC target genes.

Other comments:

1. To better understand the sequencing samples, the authors should include a supplementary table about samples they generated, the sequencing depth, and quality of sequencing data (mapped reads etc.).

2. The level of reintroduced SMARCB1 in PDO should be shown to see its relative levels as compared to normal endogenous levels in cell lines that bear SMARCB1.

3. Fig.1C only showed the changes of CTCF and RAD21 in the regions with ATAC-Seq changes, if the authors analyze the ChIP-seq of CTCF and RAD21 directly, how many peaks in total are altered?

4. Fig.1B, is the ATAC-Seq showing one of the replicate? Or merged data from replicates? The lost/gain at the bottom of Fig.1B, how are they called? The gained sites at the bottom only seem to show 4-5 peaks, but by eye many peaks in the top row of ATAC-Seq track seem to be gained peaks.

5. For Fig.1B, gene tracks should be added to better know are these noncoding regions or coding

genes.

6. Fig.2A, can RhoME3 also be included in the Hi-C map? Was the loop (circled) out only called in P130 PDO and won't be called in re-expression? Or is it quantitatively altered?

7. Only 29 ATAC-seq peaks were consistently seen in all 3 PDOs, what are the genes associated with these? Is MYC loci one of the few with consistent changes of ATAC-Seq in all 3 PDOs?

Minor:

Line 85 – references were not converted to numbered citations.

Reviewer #4:

Remarks to the Author:

The authors used datasets from various experiments including single cell multi-omics data to study the effect of SMARCB1 mutation in malignant rhabdoid tumor (MRT). I suggest that the authors address the following comments:

1. A number of experiments were performed and it would be helpful to have a table to organize the experiments and datasets, including information on technology (RNA-seq, ATAC-seq, ChIP-seq, Hi-C, qPCR, 10x multiomics, etc), sample (P78, P60, P103, P156, ...), condition (control or SMARCB1+), and figures that is related to each dataset.

2. From single cell multi-omics data, the authors found that tumor cells are different from different patients and normal cells are less variable across patients. More analysis need to be done to dive into this.

First, the UMAP shown in Fig. 5C was plotted with both the scRNA-seq modality and the scATAC-seq modality. One can plot the two modalities separately to investigate whether the same pattern holds for each modality.

Second, the authors found that enhancers RhOME1, RhOME2, and RhOME3 have different accessibility across patients. The following analysis can be performed to investigate what else cause differences between patient tumors: (1) using the scRNA-seq modality, one can perform differential expression analysis to find genes that are differentially expressed in the tumor cells between patients; (2) using the scATAC-seq modality, one can also perform differential accessibility analysis to find which regions are differentially accessible across patient tumors.

3. In line 435, please give full name for SCT and TF-IDF, and specify which package and which version was used to perform the normalization.

Rebuttal NCOMMS-23-03210-T: "SMARCB1 loss creates patient-specific MYC topologies that drive malignant rhabdoid tumor growth"

We thank the Reviewers and Editor for their constructive suggestions. In response to their comments, we have performed additional experiments and analyses, as detailed below.

REVIEWER COMMENTS

Reviewer #1, expertise in rhabdomyosarcoma and models (Remarks to the Author):

Malignant rhabdoid tumors (MRT) are aggressive pediatric malignancies characterized by bi-allelic inactivation of SMARCB1. It is remarkable that the genome of MRTs is deficient in other putative known oncogenic drivers and deletions of tumor suppressors, thus suggesting that SMARCB1 loss is the driving mutation. Despite this knowledge and apparent genomic simplicity, the survival of children with MRT is dismal. Herein, the authors use patient-derived organoids (PDO) and multi-omics approaches to define SMARCB1 loss-dependent epigenetic alterations that could provide insight into MRT tumorigenesis. Upon re-introduction of SMARCB1 into MRT PDOs, the authors found prominent looping changes surrounding the MYC promoter driving high MYC expression levels in control samples. Knockdown of MYC revealed reduced MRT PDO proliferation suggesting these alterations are essential to drive MRT cell proliferation. Analysis of additional PDOs revealed patient-specific SMARCB1 loss-dependent looping alterations connecting super enhancers with the MYC promoter. These interactions are termed Rhabdoid Oncogenic MYC Enhancers (RhOMEs) by the authors. Two of the RhOMEs were previously implicated in enhanced MYC levels in tumors while a third is newly identified. Additionally, the heterogeneous enhancer-promoter interactions were found in independent primary patient tumor samples. The authors hypothesized these aberrant topologies could be maintained by residual ncBAF and showed that inhibition of the BRD9 subunit phenocopied SMARCB1 reconstitution, hinting at a mechanism of ncBAF mediated tumorigenesis of MRT.

In total, this study provides new insights into the epigenetics of MRT implicating epigenetic regulation of MYC in SMARCB1- deficient tumors.

Specific Comments

1. There is a noticeable lack of hypothesis in the introduction. Hypotheses and support are provided throughout the results but would serve the paper well to be included in the introduction. Also in introduction lines 49-51, the authors make a claim that loss of SMARCB1 is not sufficient for MRT development. This statement is based upon SMARCB1 loss being detected in adjacent normally appearing Schwann cells in MRTs. This observation more likely reflects the lack of transformability of Schwann cells rather than the sufficiency of SMARCB1 loss driving tumorigenesis in the still to be identified cell of origin. In fact, data from a genetically engineered mouse model suggests that Smarcb1 loss is sufficient for MRT development (PMID: 12450796) and should be cited.

Answer:

As outlined by the reviewer, we previously demonstrated that Schwann cells and MRT share a common precursor from the neural crest lineage (Custers et al. 2021). We suggest that the loss of SMARCB1 in this precursor can result in MRT development, but that these cells can develop into histologically normal appearing Schwann cells as well. Conditional inactivation of SMARCB1 at a specific time window during development was previously demonstrated to induce MRT (Roberts et al. 2002; Vitte et al. 2017; Graf et al. 2022). Furthermore, we and many others have shown that there are no other recurrent genetic alterations identified in MRT besides SMARCB1 loss (Tang and Verhaak 2016; Torchia et al. 2016; Chun et al. 2019; Erkek et al. 2019). Together, these data strongly suggest that SMARCB1 loss is required, but not sufficient for MRT development and that other non-genetic (epigenetic) mechanisms are involved. We agree that loss of SMARCB1 in mature Schwann cells will most likely not transform

them into MRT, but this is not what we demonstrated in our previous publication. We now clarified this further in the Introduction section of our revised manuscript.

Changes:

(1) We changed part of the introduction (Line 51 to 57):

“Earlier studies revealed the importance of SMARCB1 loss to drive MRT formation. For instance, SMARCB1 loss at a certain time window during murine embryonic development is sufficient to initiate MRT formation⁸⁻¹⁰. We recently found that bi-allelic SMARCB1 inactivating mutations in MRT can be shared with adjacent morphologically normal Schwann cells, suggesting that loss of SMARCB1 is required but not sufficient for MRT development¹¹. No other recurrent genetic alterations have been identified in MRT^{6,12-14}, suggesting that cell type or cell state-specific epigenetic mechanisms guided by SMARCB1 loss further drive malignant transformation.”

(2) We cited Roberts et al. 2002 in the revised manuscript.

2. They state in extended 3B there are 131 loci identified. The authors should include the list of loci and potential interacting promoters. There was a lack of rationale why MYC was chosen other than its MYC. There were no comments about other interactions found besides those linking super enhancers with the MYC promoter. A general comment about the other interactions would aid in the rigor behind exploring the MYC enhancer-promotor interactions.

Answer:

We thank the reviewer for pointing this out. In our original submission, we focused on the loci surrounding the MYC gene because of the previously described role of MYC in malignant rhabdoid tumorigenesis (Chun et al. 2019; Custers et al., 2021). The interaction of the MYC promoter and RhOME 2 (E+1.1 Mb) is the number 6 most changing interaction in our Hi-C analysis (Supplementary Table 2) and - as MYC being a known driver of MRT - this motivated us to look further into this interaction. Nevertheless, we agree with the reviewer that the other interactions should not be ignored. We therefore include the complete list of identified chromatin loops upon SMARCB1 reconstitution in our revised manuscript and explain our prioritization strategy in more detail.

Changes:

(1) We added the Supplementary Table 2 describing all the differential loops and their putative gene targets (if there is a promoter involved in the interaction).

(2) Additional explanation in the Results section (Line 124 to 128):

“Ranking the most prominently lost loci upon SMARCB1 reconstitution, we found an interaction between the promoter of the MYC oncogene and a ~1.1 Mb distal region (Fig. 2A). This loop is the sixth most reduced interaction following SMARCB1 reconstitution, and the top ranked interaction involving a proto-oncogene. This distal region of the MYC promoter is marked by high H3K27ac levels indicative of a super enhancer (Fig. 2A,B and Supplementary Fig. 4B, Supplementary Table 2).”

3. The authors perform experiments attempting to delete ncBAF activity with BRD9 inhibitors in MRT. These are perplexing given that MRT has been illustrated to be dependent upon ncBAF activity. This work is cited in reference 33. The results should be presented with the context that BRD9 inhibition results in cell death in MRT.

Answer:

As shown in Fig. 6A, BRD9 inhibition induces a differentiation phenotype similar to SMARCB1 reconstitution in our models, not cell death. Indeed, Wang and colleagues suggested that in MRT cell lines (such as G401 cells) BRD9 inhibition induces cell death (Wang et al. 2019). However, the statement of cell death in the paper by Wang et al requires a more nuanced interpretation. The authors used MTT and CellTiter-Glo assays to measure viability. What is measured by these assays, however, is the metabolic activity of cells, not cell death. Performing CellTiter-Glo assays on two of our PDO models as well as G401 cells, the latter of which were also used by Wang et al, we observe the same effect as seen by Wang et al (Fig. R1). Our data supports the earlier observations made by Wang et al that inhibition of ncBAF activity inhibits proliferation of MRT cells. We further show that BRD9 inhibition induces a differentiation phenotype (Fig. 6A in our manuscript), similar to SMARCB1 reconstitution, which has not been evaluated by Wang et al. Therefore, our findings further strengthen BRD9 inhibition as a potential therapeutic vulnerability of MRT.

Fig. R1: CellTiter-Glo-assays to measure cell viability of the indicated MRT models treated with either vehicle (DMSO), I-BRD9 [10 μ M], or BI-9564 [10 μ M] for 120 hours ($n = 3$). Statistical significance was tested by unpaired t-test (*: $p \leq 0.05$, **: $p \leq 0.01$, ***: $p \leq 0.001$, ****: $p \leq 0.0001$).

Changes:

Fig. R1 was added to the manuscript as Supplementary Fig. 7A.

4. Line 212-213, the authors state that they visually inspected the data. This statement provides no benefit to scientific rigor, and actually detracts from the manuscript. The authors should consider altering this language.

Answer:

We agree that this sentence is misplaced and rephrased it as suggested by the reviewer.

Changes:

(1) We have rewritten the paragraph in the Result section (Line 232 to 234):

“We performed RNA-seq to further measure the transcriptomic changes following ncBAF inhibition and found a significant association between gene expression changes induced after I-BRD9 treatment and those upon SMARCB1 reconstitution (Fig. 6C).”

5. The data presented in this manuscript does not clearly depict the mechanism in the bottom panel of the cartoon in Figure 7. It is provocative but should be presented as such.

Answer:

This is indeed a good point and we thank the reviewer for pointing us to this discrepancy in the model. We therefore decided to remove the figure from our manuscript.

Minor Comments

1. The authors did not comment on the cell type heterogeneity in PDO samples. They performed bulk analyses on the PDO to find tumor cell specific looping interactions, but it remains unclear if PDO are a homogenous population of tumor cells. The authors single cell ATAC analysis of primary patient samples shows that the interactions likely would have been pulled out from a heterogeneous mixture, but the authors should consider addressing pitfall.

Answer:

A previous study performed by our lab (Custers et al. 2021) employed single cell RNA-sequencing to show that our PDO population is composed of a heterogeneous mix of tumor cells with different similarity scores towards different developmental stages of neural crest development. As the same three models were also used for the bulk analyses, we are confident that we are still looking at a heterogenous population of tumor cells within our PDO model.

Several of the techniques used in this paper cannot be performed at the single cell level with adequate quality/resolution. Although we cannot estimate the chromatin loop formation within single cells, we could assess chromatin accessibility of the RhOME sites at a single cell level using scMultiome. However, we found no indication that there is differential RhOME usage within subpopulations of a tumor, suggesting homogeneous usage of one or a combination of RhOMEs within the same tumor. A more detailed explanation about intratumoral heterogeneity is discussed below in response to a comment of Reviewer #2 (p. 7-8).

Changes:

We added a section to the Discussion to address this issue (Line 278 to 281):

“The use of these super enhancers is identified from a heterogeneous population of cancer cells, suggesting also distinct usage and activity of these super enhancers within one patient. The degree of heterogeneity in chromatin accessibility within one patient remains to be further investigated.”

2. References on line 84-85 need formatting.

Answer:

The reference was changed to proper formatting.

Reviewer #2, expertise in SMARCB1/MYC/rhabdoid tumours and scRNA-seq (Remarks to the Author):

In the manuscript „SMARCB1 loss creates patient-specific MYC topologies that drive malignant rhabdoid tumor growth“ Lui et al. characterize SMARCB1-dependent regulatory landscape by using patient derived organoid models. Even though the manuscript gives some novel insights into mechanisms of this rare, malignant entity, I have some major concerns, which needs to accepted before acceptance:

Major concerns

- How was lentiviral transduction done? In a PDO consisting of multiple cells? If yes, how was the transduction efficacy? Do we deal with a heterogeneous model consisting of SMARCB1 positive and SMARCB1 negative cells at the same time?

Answer:

Lentiviral transduction was performed as we previously described (Koo et al 2011; Custers et al. 2021). We now describe this procedure in more detail in the Methods section of our revised manuscript. Transduced cells were positively selected using antibiotic selection (blasticidin). After selection (i.e., when all non-transduced control cells have died), a population consisting solely of transduced cells is remaining, which is supported by 1) their blasticidin resistance and 2) a complete differentiation phenotype of the cells that are transduced with the SMARCB1 expression construct.

Changes:

We added a more detailed description of the transduction procedure to the Material and Methods section (Line 362 to 368):

“Lentiviral transductions were performed as described^{11,52}. For SMARCB1 reconstitution, MRT PDOs were transduced with pLKO.1-UbC-luciferase-blast or pLKO.1-UbC-hSMARCB1-blast lentiviruses, as described¹¹. PDOs were dissociated into single cells and transduced via spinoculation at 32°C for one hour (600 x g) with a virus MOI of ± 0.4. After four hours of recovery at 37°C, cells were re-seeded in basement membrane extract (BME). After two days, 10 µg/ml blasticidin was added to the culture medium. After selection (i.e., four days after transduction when all non-transduced control cells died), cells were harvested for the different applications.”

- Why do the authors report first about the findings in one/two PDO and later of some related findings in the other PDOs? This is confusing, especially the finding and conclusion found in these four PDOs are heterogenous. I would recommend to reconstruct the way of story telling by integrating all PDOs from the beginning.

Answer:

We understand the comment raised by this reviewer and have considered the order proposed above. However, we decided to first present the data in one PDO and subsequently use the other models to validate our findings and to introduce the observed patient specific effects, which is a key finding of our manuscript. Since none of the other reviewers nor the editor considered this to be a pitfall, we decided to keep the order as it was in our original submission.

- As stated before the authors report about heterogeneous findings in four PDOs. Even though they validate some findings in patient samples, the questions remains, if it is not possible to increase the number of PDOs for these analyses or at least for parts of these analyzes.

Answer:

MRT is an exceptionally rare cancer. In our center (Princess Maxima Center, the largest pediatric cancer center in Europe), we would expect to treat no more than two to three cases per year. Many of these children will only be treated with chemotherapy without receiving surgery. Therefore, tissue availability for establishing PDOs is extremely limited, which makes it impractical to increase the number of PDOs for the increment analysis as suggested by the reviewer within a relevant time frame for this manuscript. Despite having studied only three cases, our results are remarkably reproducible in the n=7 patient tissues, which is a large cohort for this rare entity, to which we applied scMultiome sequencing.

- The author should indicate the number of samples used in this study (e. g. publically available ATAC seq samples, which were reanalyzed).

Changes:

We now include a table (Supplementary Table 3) in our revised manuscript summarizing the publicly available samples used in this study.

- I recommend to validate findings of the regulator landscape detected in MRT PDO in patients samples also on other levels, e. g. by including ChIP-seq of CTCF and Hi-C.

Answer:

We agree with the reviewer that this would be a nice addition to the study. However, as outlined above, tissue availability is limited because of the rarity of MRT and we mostly receive leftover material from needle biopsies after intense therapeutic regimens, limiting high throughput approaches on tissue. The input amount needed for high quality ChIP-seq or Hi-C is around $5 - 10 \times 10^6$ of viable cells. Acquiring sufficient numbers of cells to perform scMultiome, ChIP-seq and HiC is simply not feasible. Also, our experience is that it is technically challenging or impossible to generate Hi-C data with sufficient quality from frozen materials. We therefore prioritized scMultiome as it allows for getting two readouts simultaneously thereby making optimal use of the limited material.

- Seven MRT patient samples were used for scRNA-seq, as well as scATAC seq. If I understand correct also in patient sample intertumoral heterogeneity for OCR at RhOME1-3 can be identified. How is this heterogeneity within one tumor (intratumoral heterogeneity)?

Answer:

Motivated by this interesting suggestion, we performed several additional analyses on our scMultiome data. It is indeed correct that the patient samples analyzed display different OCRs at RhOME 1-3, thereby demonstrating intertumoral heterogeneity on the level of enhancer usage. Addressing differences between individual cells within the same tumor, intratumoral heterogeneity, turned out to be challenging. We inspected the patient samples having the highest chromatin accessibility at the identified RhOMEs (P052, P041 and P156) in more detail by performing intratumoral sub clustering (included only in reply to the reviewer as Fig. R2). The obtained single cell ATAC-seq data, however, is rather sparse with a relatively low genome-wide coverage. As such, in less than 6% of the cells within each tumor cell cluster reads could be detected covering the different RhOMEs (Fig. R2A). Within the cells having reads, we find that all cells seem to have preferential use of (a combination of) RhOME(s). In some cells, however, we detect some reads at other RhOMEs as well, which could point towards intratumoral heterogeneity (Fig. R2B). But since the coverage is too low, we do not feel confident enough to make such a statement.

Fig. R2: (A) Dot Plot depicting average delta.counts (from the scATAC data) at the indicated RhOME regions for the different patient tumors. (B) Individual clustering of three patient tumors, grouping variables based on joined clustering. Stacked bar graph depicts accessibility of one or a combination of RhOMEs in the individual tumor subclusters.

Changes:

We have added a section to the Discussion section covering the absence/ presence of intratumor heterogeneity (Line 278 to 281):

“The use of these super enhancers is identified from a heterogeneous population of cancer cells, suggesting also distinct usage and activity of these super enhancers within one patient. The degree of heterogeneity in chromatin accessibility within one patient remains to be further investigated.”

- Can the findings of heterogeneous regulator landscape be mechanistically explained? Or are there correlations to genetics (type of SMARCB1 alteration) or clinical differences (e. g. localisation of tumor occurrence)?

Answer:

We agree that the correlation analysis will be interesting. However, the number of patient samples are too low to perform such calculations. Due to the rarity of MRT, it is currently simply not feasible to perform such analysis and we therefore consider this outside the scope of our manuscript. We speculate, however, that the patient-specific use of enhancers in MRT reflects the timing at which the tumor originated during neural crest development. Neural crest development is a highly dynamic process and chromatin looping and chromatin organization will change dramatically during cell state changes. Thus, a certain epigenetic cell state might be accompanied by certain interactions that contribute to tumor initiation (Latil et al. 2017; Roe et al. 2017). Alternatively, it could also be that certain clones were selected based on competitive advantage based on the (in)activity of certain enhancers. We discuss this in the Discussion section of our manuscript (Line 270 to 275):

“The patient-specific enhancer landscapes found in MRT could reflect the developmental origin of MRT, which lies in the neural crest¹¹. Neural crest development is characterized by rapid switching of cell states caused by, amongst others, chromatin reorganization to assure quick and simultaneous

development of several different cell types⁴⁵⁻⁴⁷. We hypothesize that loss of SMARCB1 in a specific cellular context during neural crest development prevents the inactivation of certain MYC enhancers, which is essential for proper lineage specification.”

Reviewer #3, expertise in Hi-C/4C, ATAC-seq, ChIP-seq analysis (Remarks to the Author):

In Liu et al. entitled: “SMARCB1 loss creates patient-specific MYC topologies that drive malignant rhabdoid tumor growth”, the authors utilized malignant rhabdoid tumor (MRT) patient-derived organoid (PDO) models to study how biallelic loss of SMARCB1 may impact gene expression, epigenome, and 3D chromatin topology in MRT. Indeed, SMARCB1 is deleted in >95% of MRT, and it is important to understand how its loss causes cancer. The authors conducted re-expression of SMARCB1 in PDO model (generating a line they called SMARCB1+) and compared it to MRT PDO cells without this protein (WT). They utilized a series of epigenomics assays, including ATAC-Seq, ChIP-Seq, RNA-Seq, Hi-C, 4C, and found that SMARCB1 re-expression globally alters the enhancer landscape (more gain than loss); by Hi-C, they found that while TADs were largely kept, there was chromatin topology difference in the MYC gene loci. The authors further showed that MYC may be activated by different enhancers in a patient-specific manner in different MRT patients; and tried to explore this idea with single-cell multi-omics data from patient’s tumors. Finally, they attempted a therapeutic avenue for MRT by a pharmacological inhibitor of BRD9.

Overall, this paper was well-written with a clear description of the background and methods. Genomics experiments and analyses were generally performed well. However, I found this paper does not have a central point to make; for several potential points the paper seems to make, the novelty is not strong (see below); it is unclear if any epigenome or chromatin topology change observed is due to direct or secondary effects of SMARCB1 loss/gain; the mechanisms of such changes are also unclear; the main message in the title about patient specific MYC chromatin topology is not fully supported by the data and appeared pre-mature; the treatment by iBRD9 while interesting is not particularly new (and whether the drug is specific to BRD9 or how does it work are unclear, i.e., degrading BRD9 or preventing its chromatin binding).

Main concerns:

1. Novelty issue:

a. The re-expression of SMARCB1 into its null tumor cells to examine epigenomic changes has been conducted by a few studies (Wang et al., 2017, PMID: 27941797; Nakayama et al., 2017, PMID: 28945250). The novelty of this paper is partially compromised. One potential remaining novelty is that this paper used Organoid system to achieve this goal.

Answer:

We do not agree with the reviewer that there is a novelty issue. The fact that we use MRT organoid models from different patients in our study allowed us to describe for the first time that MRT development can be driven by patient-specific epigenetic reprogramming caused by SMARCB1 loss. It is widely accepted that cancer cell lines in general are poorly representative of patient tumors ((Masters 2000; Ho et al. 2020). Furthermore, only a handful of MRT *in vitro* models are available, therefore not allowing for the investigation of patient-specific tumor driving mechanisms. We previously demonstrated that MRT organoids are representative of patient tumors on the genetic, epigenetic and gene expression level (Calandrini et al. 2020). MRT PDOs can be established at very high efficiency from patient tissues, allowing us to study such patient-specific mechanisms. So, although similar *SMARCB1* re-expression experiments were indeed performed in MRT cell lines before, we for the first time report on the patient-specific epigenetic landscapes driving MRT growth.

b. The epigenomic changes of histone marks have been examined previously after SMARCB1 re-

expression, even Hi-C was also conducted by Nakayama et al., 2017, PMID: 28945250, although the authors suggested limited changes therein.

Answer:

The Hi-C described by Nakayama et al. 2017 was performed on a cancer cell line (VA-ES-BJ) derived from an epithelioid sarcoma. Although SMARCB1 deficient too, this is an entirely different tumor entity derived from a completely different embryonic lineage (mesoderm, not neural crest). It is widely accepted that epigenetic gene regulation in particular is highly cell type and lineage specific and therefore also very likely to be different in different tumor types (Plass et al. 2013; Klughammer et al. 2018). The observation by Nakayama et al, that SMARCB1 reconstitution did not significantly change the 3D organization at the *MYC* locus (Fig. R3, only included in the reply to the reviewer's comment), in this single epithelioid sarcoma cell line, confirms this further. Therefore, we show for the first time the existence of patient-specific chromatin loops that control expression of important oncogenes (intertumoral epigenetic heterogeneity).

Fig. R3: Contact map of the MYC locus using Hi-C data from Nakayama et al. (2017) showing that genome contacts are not changing at the MYC locus upon SMARCB1 re-expression in VA-ES-BJ cells. H3K27Ac ChIP-seq also does not show a significant decrease at the MYC gene, implying that MYC expression is not regulated by SMARCB1 in this cell line.

c. The role of BRD9 in helping BAF aberrant chromatin location upon SMARCB1 loss in MRT was reported elsewhere (Wang 2019, PMID: 31015438). Wang et al., not only identified BRD9 as required for MRT growth, but also showed that bromodomain inhibitors of BRD9 did not affect cell growth, implying that BRD9 degradation rather than domain inhibition is required for cell inhibition in MRT. This current paper merely treated PDO with BRD9 inhibitor (a different iBRD9, which may have off targeting to BRD7, see MedChemExpress, HY-18975 information).

Answer:

Thanks for pointing this out to us. Motivated by this comment, we treated two of our MRT PDO models as well as G401 cells (used in the Wang et al study) with I-BRD9 and one of the BRD9 inhibitors used by Wang and colleagues, BI-9564. We observe that treatment of all three cell models with I-BRD9 as well as BI-9564 significantly reduces cell growth (Fig. R4A), strongly suggesting that the observed effects are not caused by an off-target effect. Furthermore, we performed CUT&RUN for BRD9 on MRT PDOs that were treated with I-BRD9 revealing a drastic decrease in binding of both BRD9 and SS18 to the *MYC* locus (Fig. R4B), demonstrating the direct effect of BRD9 inhibition on ncBAF binding.

Fig. R4: (A) CellTiter-Glo-assays to measure cell viability of the indicated MRT models treated with either vehicle (DMSO), I-BRD9 [10 μ M], or BI-9564 [10 μ M] for 120 hours ($n = 3$). Statistical significance was tested by unpaired students t-test (*: $p \leq 0.05$, **: $p \leq 0.01$, ***: $p \leq 0.001$, ****: $p \leq 0.0001$). (B) Chromatin accessibility (as measured by ATAC-seq), BRD9 and SS18 binding at the MYC locus in P103.

Changes:

(1) Fig. R4A was added as Supplementary Fig. 7A in the revised manuscript and Fig. R4B as Fig. 6F.

(2) We have added the following sentence to the Result section of the revised manuscript (Line 229 to 230):

“We observed that, morphologically, MRT cells exhibited a differentiation phenotype similar to SMARCB1 reconstitution¹¹ (Fig. 6A) and cell growth was significantly inhibited (Supplementary Fig. 7A).”

d. This paper conducted Hi-C and found chromatin changes in MYC loci (which may be considered a potential novelty, although global analysis needs some clarification, see below). Unfortunately, the authors provided no explanation of mechanisms of this finding.

Answer:

We agree that the mechanism causing the intertumoral heterogeneity is very interesting, but so far we have not been able to find an explanation for these exciting observations. Amongst others, we used whole genome sequencing to analyze the DNA sequence at the identified RhOMEs, but could not find any differences between patients that could explain differential enhancer usage. We also investigated the DNA methylation status at the RhOMEs but again no differences could be found. The most likely explanation, to our opinion, is that the observed patient specificity is reflecting the epigenetic state of the cell of origin (for MRT this is a neural crest progenitor) at the moment it became malignant. This is supported by preliminary analyses showing that different cell types within the neural lineage as well as neural crest cells depict activity of the enhancers (Fig. R5, only included as response to the reviewers). The H3K27Ac signal suggests that these ‘normal’ enhancer regions did not yet function as super

enhancers, therefore we hypothesize that at the moment of tumor initiation a switch is made towards a ‘super enhancer’ activity. Again, these analyses are preliminary and we therefore decided to only include them in the reply to the reviewer’s comment (Fig. R5). Further research is required to elucidate this in more detail, but we consider this outside the scope of our current manuscript. We do however discuss it in more detail in the Discussion section of our revised manuscript (Line 270 to 275):

“The patient-specific enhancer landscapes found in MRT could reflect the developmental origin of MRT, which lies in the neural crest¹¹. Neural crest development is characterized by rapid switching of cell states caused by, amongst others, chromatin reorganization to assure quick and simultaneous development of several different cell types^{45–47}. We hypothesize that loss of SMARCB1 in a specific cellular context during neural crest development prevents the inactivation of certain MYC enhancers, which is essential for proper lineage specification.”

Fig. R5: H3K27Ac signal at the genomic location of RhOME2. Left panel depicts ChIP-seq signals in PDO P103 and different publicly available MRT tissues (Wang et al. 2017). Right panel depicts corresponding ChIP-seq data of several cell lines of neural and embryonal origin (Rada-Iglesias et al. 2011; Dunham et al. 2012; W. Liu et al. 2013; Prescott et al. 2015; Boeva et al. 2017).

e. This paper used an eye-catching title of patient-specific MYC chromatin topology, but this was only directly showed by Hi-C or 4C in two patient samples (the single cell data will not directly support chromatin topology, at least not at the 3D genome level).

Answer:

MRT is an extremely aggressive but rare cancer. Between our centers, (Netherlands Cancer Institute and the Princess Máxima Center - the largest pediatric cancer center in Europe), we would expect to treat no more than two to three patients per year with MRT. Many of these children will progress under intense chemotherapy treatments and not undergo surgery making access to tissue limited. The number of models and samples in our study is in fact one of the largest for MRT (for scMultiome the only) and therefore a significant contribution to the field. Although scMultiome is not a measure of chromatin topology, it does, in combination with the *in vitro* data, provide strong evidence for the patient-specific use of these enhancers in tumor samples.

Overall, this paper put in a lot of points that are seemingly novel, but careful inspection of each of these indicates that the novelty is moderate (mostly come from the PDO system, and Hi-C helps somewhat), and some datasets, (e.g., patient 4C/Hi-C and single cell multi-omics data (as resource).

Answer:

The strength of our manuscript lies in the combination of patient-specific *in vitro* model systems with a set of different technologies, covering multiple areas of the molecular characterization of MRT, where novel conceptual insights into the mechanisms driving tumorigenesis are desperately needed. This led us to the discovery of a novel concept of patient-specific oncogenic activation via an epigenetic event.. Our findings therefore can act as a blueprint to unravel the contribution of intertumoral epigenetic heterogeneity to the development of other cancers.

2. Lack of mechanism: it is unclear if any epigenome or 3D chromatin topology change observed in PDO after SMARCB1 re-expression is due to direct or secondary effects of SMARCB1.

Answer:

Although we agree that it is interesting to investigate whether the observed effects are direct or indirect, in our opinion this is not of relevance for our current study. The observation that MRTs can be reverted to a normal phenotype just by reconstitution of SMARCB1 expression gives us the opportunity to find tumor-driving mechanisms downstream of SMARCB1 that could be therapeutically exploited. Whether these mechanisms are directly or indirectly regulated by SMARCB1 is not the focus of our manuscript. Nevertheless, in our revised manuscript we perform ChIP-seq for SMARCB1 demonstrating that, upon reconstitution, SMARCB1 is bound at SMARCB1+ specific OCRs, suggesting a direct effect at least on these regions. The results of this ChIP-seq are further discussed in the reply of the reviewer's next question.

Changes:

We included the SMARCB1 ChIP-seq in Fig. 2B and Supplementary 2A and B of the revised manuscript.

a. The authors need to show where SMARCB1 or other SWI/SNF complex bind in the genome after SMARCB1 re-expression (this is commonly done by many previous papers, Wang et al., 2017, Nakayama et al., 2017). They need to correlate the possible gained/lost binding of SWI/SNF with the altered epigenome in Fig. 1/2/3. At this stage, these are just phenomenon, it is completely unclear how SMARCB1 plays the role directly.

Answer:

This is a great suggestion. Motivated by it, we performed CUT&RUN in P103 with and without SMARCB1 reconstitution on BRD9 (ncBAF), SS18 (cBAF and ncBAF), and ChIP-seq on SMARCB1. These experiments confirmed that SMARCB1+-specific enhancers (gained OCRs) are bound by SMARCB1 and SS18 suggesting that SMARCB1, as part of the cBAF complex, directly regulating their activity (Fig. R6A)). We then stratified the 3 k-mean clusters (Fig. 4A in the manuscript) of the control-specific OCRs (lost upon SMARCB1 re-expression). Interestingly, this revealed that SMARCB1 binding is most gained on OCRs in cluster K3, which is the cluster containing the patient-specific enhancers, while binding of BRD9 and SS18 is markedly reduced (Fig. R6B). These results indicate that gene regulation in MRT cells is highly dependent on binding of the ncBAF complex. Reconstitution of SMARCB1 reduced the occupancy of the ncBAF complex at these loci and is therefore required to prevent the interaction of a distal super enhancer with gene promoter (like *MYC*) through chromatin looping.

Fig. R6: Chromatin occupancy of SMARCB1, BRD9, and SS18 at the differential open chromatin sites in the control and SMARCB1+ P103 organoids. (A) Binding of SMARCB1, BRD9 and SS18 at gained and lost OCRs as measured by ATAC-seq (Fig. 1C in the manuscript) and (B) at 3 k-means clusters of the lost chromatin sites as shown in Supplementary Fig. 5A; (C) Chromatin occupancy of the indicated proteins at the two example loci as shown in Fig. 1B of the manuscript.

Changes:

(1) We included the tornado plots of SMARCB1, BRD9 and SS18 in Fig. R6A in Supplementary Fig. 2A, R6B as Supplementary Fig. 5A and Fig. R6C as Supplementary Fig. 2B.

(2) We have added the following sections discussing these experiments in the manuscript:

Result section (Line 81-94):

“To find these tumor-driving regulatory changes, we lentivirally transduced a MRT PDO model (named P103)²³ with either a Luciferase expression (Control) or a SMARCB1 expression (SMARCB1+) plasmid and measured chromatin accessibility by assay for transposase-accessible chromatin using sequencing (ATAC-seq) and BAF chromatin occupancy by chromatin immunoprecipitation sequencing (ChIP-seq) or Cleavage Under Targets & Release Using Nuclease sequencing (CUT&RUN) (Fig. 1A,B, Supplementary Fig. 2A). Following SMARCB1 reconstitution, we found 7,941 newly formed open chromatin regions (OCRs) that are enriched for transcription factor motifs from different families such as SMARCC1/2 and AP-1²⁴ (Fig. 1C, Supplementary Fig. 1B,C). SMARCB1 ChIP-seq revealed that these OCRs are bound by SMARCB1 (cBAF and PBAF) and SS18 (ncBAF and cBAF), indicating cBAF binding at these regions upon reconstitution (Supplementary Fig. 2B). When we performed functional annotation of these OCRs using GREAT, we found that several categories are enriched, mostly related to differentiation and developmental processes (Supplementary Fig. 1D). At the 1,211 OCRs that were lost, an apparent decrease in binding was observed of the ncBAF complex members BRD9 and SS18 (Supplementary Fig. 2B).”

Result section (Line 137 to 143):

“To explore whether ncBAF complex binding was affected by SMARCB1 reconstitution, we performed CUT&RUN on BRD9 (ncBAF) and SS18 (cBAF and ncBAF). We found that both BRD9 and SS18 binding at the MYC promoter as well as at the 1.1Mb distal region is dramatically reduced upon SMARCB1 reconstitution (Fig. 2B). These results indicate that MYC expression is at least for a large part dependent on binding of the ncBAF complex. Reconstitution of SMARCB1 reduces ncBAF complex binding at the MYC locus thereby likely inhibiting the interaction of the distal super enhancer with the MYC promoter.”

Discussion (Line 316 to 317):

“Therefore, MRT initiation during development might be dependent on the binding at, and activity of the ncBAF complex at chromatin regions regulating oncogene expression thereby driving tumorigenesis ”

b. Many of these changes are likely indirect effects/secondary effects after long term SMARCB1 re-expression. A possible strategy is to re-express SMARCB1 with a FKBP12 tag so it can quickly induce or reduce by dTAG to achieve rapid regulation to understand direct roles.

Answer:

The suggestion of the reviewer is very interesting and would indeed make a nice follow-up of the current study. As described in the reply to comment 2 of the reviewer, however, we feel that whether the observed effects are direct or indirect is not of relevance for our current study. Nevertheless, our SMARCB1 ChIP-seq results strongly suggest that most of the observed effects are direct.

c. The reasons for changes of the MYC loci Hi-C are also unclear. Did SWI/SNF bind any of these enhancers/promoters directly after re-introduction and then directly suppress MYC? Or did SWI/SNF now gains their “correct” location after SMARCB1 re-introduction, and thus MYC loses SWI/SNF binding at enhancers.

Answer:

This is an important point. To investigate this in more detail, we performed CUT&RUN experiments on BRD9 and SS18 and double-crosslinked ChIP-seq on SMARCB1 in MRT PDOs with and without

SMARCB1 re-expression. We observed increased binding of SMARCB1 after reconstitution at both the ATAC-seq peaks identified in the control and SMARCB1+ condition (Fig. R7A, B, only included in the response to the reviewers), suggesting that the loci indeed re-gained SMARCB1 binding upon reconstitution. SMARCB1 binding is strongly increased in SMARCB1+ specific OCRs, while BRD9 binding (ncBAF subunit) is reduced at these loci upon SMARCB1 reconstitution (Fig. R7A,B). The overall chromatin occupancy of BAF and ncBAF, indicated by the binding of the shared subunit SS18 (Fig. R7A,B), remains largely constant. We then evaluated the binding of these three subunits at the *MYC* locus before and after SMARCB1 reconstitution. While BRD9 and SS18 binding dramatically decreased at the *MYC* promoter after SMARCB1 reconstitution (Fig. R7C), no increased binding of SMARCB1 was observed. This data indicate that downregulation of *MYC* expression after SMARCB1 reconstitution is a consequence of the loss of ncBAF complex binding, rather than the cBAF complex 'hijacking' the binding sites of ncBAF.

Fig. R7: Genome-wide and locus specific binding of SMARCB1, BRD9, and SS18 in the control and SMARCB1+ P103 organoids. Binding of SMARCB1, BRD9 and SS18 at the peaks called in CTRL (A) and SMARCB1+ (B) samples, respectively; (C) binding of SWI/SNF and other factors at the MYC locus.

Changes:

(1) We have included Fig. R7C as updated Fig. 2B in the manuscript.

(2) We have added following section to the result section of the manuscript (Line 137 to 143) describing the results of the CUT&RUN after SMARCB1 reconstitution:

“To explore whether ncBAF complex binding was affected by SMARCB1 reconstitution, we performed CUT&RUN on BRD9 (ncBAF) and SS18 (cBAF and ncBAF). We found that both BRD9 and SS18 binding at the MYC promoter as well as at the 1.1Mb distal region is dramatically reduced upon SMARCB1 reconstitution (Fig. 2B). These results indicate that MYC expression is at least for a large part dependent on binding of the ncBAF complex. Reconstitution of SMARCB1 reduces ncBAF complex binding at the MYC locus thereby likely inhibiting the interaction of the distal super enhancer with the MYC promoter.”

(3) We have added the following section to the Discussion section of the importance of ncBAF binding for tumor progression (Line 296 to 299):

“Furthermore, our data suggest that the loss of SMARCB1 in MRT causes increased binding of the ncBAF complex at super enhancers. This does not seem to be caused by binding by other BAF complexes, but rather driven by a shifted balance of BAF complex compositions in a SMARCB1 proficient or deficient context.”

d. How did the 3D genome change? Did CTCF get directly suppressed by SMARCB1 reexpression at MYC loci?

Answer:

CTCF binding is reduced after SMARCB1 reconstitution (Fig. R7C and R8). Therefore, one possibility is that SMARCB1 competes with BRD9 at different RhOMEs and suppresses the binding of CTCF. Consequently, the loops are disrupted. Notably, CTCF-mediated 3D topology changes may be restricted to the MYC locus, since we did not observe clear changes of insulation at CTCF binding sites genome wide (Fig. 4B,C). Another possible mechanism is that the changes of these loops at the MYC locus are caused by CTCF-independent cohesin. We have previously shown that CTCF-independent cohesin is highly enriched at super enhancers and plays a critical role in regulating enhancer/promoter loops (Liu et al. 2021). We have also shown that open chromatin is required for the high occupancy of cohesin at super enhancers, which may act as preferential loading sites. Since SMARCB1 inactivation creates tumor specific super enhancers including RhOMEs, we speculate that they act as extrusion platforms that bring the RhOMEs in close proximity to the MYC promoter. We now briefly discuss this in the revised Discussion section.

Fig. R8: Characterization of chromatin interaction changes at the MYC locus. CTCF binding at three RhOMEs and the MYC promoter is largely reduced after SMARCB1 reconstitution. A zoom-in figure can be found in Fig. R6C.

Changes:

We added the following to the Discussion section of the manuscript (Line 309-312):

“One possible explanation for the loss of chromatin loop formation at the MYC locus is a decreased binding of CTCF-independent cohesin, which plays a critical role in regulating promoter/ enhancer loops¹. We indeed observed, high binding of cohesin at RhOMEs, possibly creating a loading site.”

3. About MYC alteration: Weissmiller et al., 2019 (PMID: 31043611) showed that SMARCB1 re-expression in 293T cells did not obviously impact MYC expression level (Western), but can antagonize MYC binding to chromatin. But this current paper showed that SMARCB1 re-expression strongly reduced MYC mRNA expression (Fig.3C) in PDO. Is this phenomenon a PDO specific one? Can the authors examine other papers that did SMARCB1 re-expression and see how MYC mRNA and/or protein changes?

Answer:

Weissmiller and colleagues performed SMARCB1 re-expression in HEK293T cells, which is an immortalized human embryonic kidney cell line. First of all, the effects of mutations in transcription factors and epigenome regulators are known to be highly cell type specific. Second, and more importantly, HEK293T cells do not have inactivating genetic alterations in *SMARCB1*. Studying the effects of SMARCB1 re-expression in a cell line that is proficient for SMARCB1 simply does not resemble SMARCB1 reconstitution in a SMARCB1 deficient cell model representative of MRT.

To exclude that the observed effects are PDO specific, we investigated the effects of SMARCB1 reconstitution on *MYC* expression in an earlier published RNAseq dataset using MRT cell lines in which SMARCB1 was re-expressed (Wang et al. 2017). In this dataset, *MYC* expression is also significantly reduced in five of the six MRT cell lines upon SMARCB1 re-expression, further validating our observations seen in MRT PDOs (Fig. R9A, only included as a response to the reviewers). Furthermore, we performed western blot for MYC protein expression in our PDOs as well as in G401 cells +/- SMARCB1 re-expression again confirming MYC downregulation also on the protein level in PDOs and cell lines upon SMARCB1 re-expression (Fig. R9B). These analyses exclude the possibility that the presented effects on MYC expression are PDO specific.

Fig. R9: MYC expression before and after SMARCB1 reconstitution. (A) Wang et al. RNA-seq dataset depicting MYC expression in six different MRT cell lines upon SMARCB1 re-expression ($n = 3$ for BT12, BT16, G402, TM87, and TTC549, $n = 2$ for G401) (Wang et al. 2017). Statistical significance was tested by unpaired students *t*-test (*: $p \leq 0.05$, **: $p \leq 0.01$, ***: $p \leq 0.001$, ****: $p \leq 0.0001$).; (B) Western Blot depicting MYC protein levels \pm SMARCB1 in P60, P103 and G401. GAPDH was used as loading control.

Changes:

We have added the western blot results (Fig. R9B) as Supplementary Fig. 4E in our revised manuscript.

4. Ext Data Fig.2A, can the authors show the P(s) curve as commonly used control/treatment two P(s) curves in addition to the log2 fold change curve? The extremely long range curves (40-100Mb) seems to increase a lot. But due to their low frequency in P(s) curve, this may be very little, so P(s) curve is helpful to see.

Ext Data Fig.2D, what is the blue difference/subtraction plot show? Fold change or subtraction?

Answer:

Thank you for this suggestion. Below, we show the relative contact probability of both CTRL and SMARCB1+ conditions (Fig. R10, included only as a response to the reviewer). Although seemingly small, there is an increase in the number of contacts in the range of 40 and 100 Mb upon SMARCB1 reconstitution.

The blue difference plot in Supplementary Fig. 2D (now Supplementary Fig. 3D) indeed depicts the difference (subtraction) between the other two ATA plots. We now clarify this in the figure legends of the revised manuscript.

Fig. R10: Relative contact probably (RCP) of the Hi-C data from CTRL and SMARCB1+ P103 organoids.

Changes

We have added a sentence to the legend of Supplementary Fig. 3D to make the panel more clear:

“(D) Aggregate TAD analysis (ATA) suggests that SMARCB1 reconstitution only has weak effects on chromatin contacts within TADs. Difference plot shows the subtraction of the Control versus SMARCB1+ ATA plot.”

5. Patient specific MYC topology.

a. The authors conducted Hi-C in one PDO, and 4C-Seq in 2 PDOs. This is too limited data to support the title: patient-specific MYC topologies. Also do the authors mean chromatin 3D topology in the title?

Answer:

Even though we only inspected three patients, the data convincingly shows patient specificity, which we confirmed in seven patient tissues using scMultiome. It is very difficult to include more PDO samples due to the rarity of the tumor entity. We re-evaluated our wording in the title and changed it accordingly (see reply to next comment).

b. Also about the title, the data did not support that “SMARCB1 loss creates patient-specific...”. These topologies are more likely due to patient variation, rather than “SMARCB1 loss creates them”. Fig.3A shows this well as even with SMARCB1 re-expression the topology still appears different.

Answer:

We apologize for this confusion and agree that the original title can be misleading and have changed it accordingly. Our data indeed did not support the conclusion that SMARCB1 loss itself creates the patient specificity.

Changes:

We changed the title of our manuscript to:

“SMARCB1 loss activates patient-specific distal MYC enhancers that drive malignant rhabdoid tumor growth.”

c. There is very strong variation of chromatin opening between patients' PDOs – e.g., line 158, “81.6%, 72.1% and 78.7% of the OCRs lost in P103, P78 and P60, respectively, are lost specifically in that

PDO". Then it becomes confusing what is the purpose of the K means clustering of ATAC-Seq from 3 PDOs of the changes of P103 (Fig.4)? and to compare to Hi-C seen in P103?

Answer:

The two analyses are very complementary. The K-means clustering was performed to investigate whether changes observed within one tumor entity can also be extrapolated towards the other two PDO lines despite the strong patient specificity. The Venn diagram was based on a stringent cutoff (FDR<0.05 and fold change ≥ 2). Every peak that did not pass this cutoff was considered as patient specific. The k-means clustering and heatmap show more quantitative patterns, but also suggest that the strongest enhancers show patient specificity. To prevent confusion, we decided to remove the Venn diagram from the revised manuscript.

Changes:

(1) We removed the Venn diagram from the manuscript and only include the k-means clustering as a more quantitative measure to explore patient specific patterns.

(2) We replaced Fig. 4A with Supplementary Fig. 4A (now 5A). A new Supplementary Fig 5A was added with additional tornado plots from the new CUT&RUN data of BRD9 and SS18 (in the rebuttal referred to as Fig. R6B).

6. The authors then moved to use single cell multi-omics in MRT to make a point that different MRT patients cells may bear different activity of the 3 enhancers close to MYC. Is this a surprise? For example, Corces M et al., 2018, PMID: 30361341 has published ATAC-Seq in hundreds of tumors. If the authors simply select some cancer samples with MYC high expression, is it uncommon that each patient use different ATAC-Seq enhancer activities? Even if so, why is it important biologically or clinically?

Answer:

Understanding how normal cells become malignant is crucial for the development of new therapies. The *MYC* oncogene is overexpressed in a majority of human cancers, which typically has a genetic cause (e.g., by an amplification of the *MYC* gene locus). *MYC* expression is known to be increased in MRT (Chun et al., 2019; Custers et al., 2021). Interestingly, MRTs lack genetic alterations of the *MYC* locus. We now provide for the first time evidence of how these tumors express high levels of *MYC*, by showing that *MYC* overexpression is caused by the accessibility and activity of patient-specific super enhancers. The epigenetic regulation of *MYC* is important to be described here, to introduce the concept of epigenetic oncogenic regulation in a patient dependent manner. Several other tumor entities, especially with little known genetic drivers, might utilize similar pro-tumorigenic epigenetic events. This may ultimately pave the path for the development of more specific epigenetic drugs in anti-cancer therapies.

Changes:

We have added the following sentences to the new Discussion to address the clinical relevance of our findings (Line 284 to 288):

"We present one of the first examples highlighting how MYC overexpression can be explained by accessibility and activity of patient-specific super enhancers. Although we only exemplify this in MRT, the concept of patient-specific epigenetic regulation of oncogenic drivers may be applicable to a broader range of tumor entities and pave the path toward more specific epigenetic drugs in cancer treatment."

7. BRD9 inhibition: the effects on gene expression and cell differentiation are very good. However while Wang et al., (PMID: 31015438) identified BRD9 as required for MRT growth, they also showed that bromodomain inhibitors of BRD9 did not seem to affect cell growth, implying that BRD9 degradation rather than bromodomain inhibition is required for cell inhibition in MRT. This current paper used i-BRD9, what is the rationale for this? Did the authors examine the drugs used by Wang et al. 2019? Did iBRD9 reduce BRD9 protein level, how about BRD7 or BRD4 levels? Is iBRD9 specific to BRD9 or also other bromodomain proteins? These need to be checked to conclude if the effect is solely achieved via BRD9 inhibition.

Answer:

To address this, we set out to test the effect of I-BRD9 and one of the BRD9 inhibitors used by Wang and colleagues, BI9564, in two of our MRT PDO models as well as in G401 cells, an MRT cell line used in the Wang et al study. Notably, as both inhibitors are chemical probes binding BRD9 at different pockets thereby inhibiting its activity, we do not expect protein levels to go down (Martin et al. 2016, Theodoulou et al. 2016, Hui et al. 2018), which we confirmed by western blot (not shown). Both inhibitors induced a significant decrease in cell proliferation in all tested cell models (Fig. R11A, included in our revised manuscript as Supplementary Fig. 7A). Furthermore, to study the effect of BRD9 inhibition on its activity in ncBAF, we measured genome-wide BRD9 binding using CUT&RUN on MRT PDOs with and without I-BRD9 treatment. This demonstrated that BRD9 binding is drastically decreased genome wide, including the *MYC* locus suggesting that observed decrease in *MYC* expression is caused by loss of ncBAF complex binding (Fig. R11B, a more extensive analysis is included in the revised manuscript in response to the reviewer's next comment).

Fig. R11: A) CellTiter-Glo-assays to measure cell viability of the indicated MRT models treated with either vehicle (DMSO), I-BRD9 [10 μM], or BI-9564 [10 μM] for 120 hours (n = 3). Statistical significance was tested by unpaired t-test (: p <= 0.05, **: p <= 0.01, ***: p <= 0.001, ****: p <= 0.0001). (B) Chromatin accessibility and BRD9, SS18 binding at the MYC locus, comparing DMSO (Ctrl) versus iBRD9 treatment.*

For BRD9 inhibitor, if the authors want to conclude that it works by preventing residual non-canonical ncBAF complex in MRT tumor cells (with SMARCB1 loss); they need to show CHIP-Seq of SWI/SNF with/without iBRD9, and how they correlate with MYC gene inhibition and the inhibition of MYC target genes.

Answer:

Thank you. As suggested by the reviewer, we performed CUT&RUN on BRD9 (ncBAF) and SS18 (cBAF and ncBAF) before (DMSO) and after I-BRD9 treatment. First, we confirmed that I-BRD9 treatment caused a global decrease in BRD9 and SS18 occupancy at BRD9 binding sites identified from untreated P103 PDOs (Fig. R12A). Next, we plotted OCRs (defined by ATAC-seq peaks) and BRD9 and SS18 peaks with and without I-BRD9 treatment revealing that, decreased accessibility co-occurs with loss of BRD9 and SS18 binding at the control-specific peaks (Fig. R12B). This strongly suggests that the active chromatin states at these loci are regulated by ncBAF complex binding. As expected, the gained peaks (SMARCB1+ specific) remain inaccessible and unbound by BRD9 and SS18 after treatment with iBRD9 (Fig. R12B). At the *MYC* locus the same switch after iBRD9 is observed; chromatin accessibility goes down together with a drastic decrease in BRD9 and SS18 binding (Fig. R12C). We included these new data to our revised manuscript as follows:

Changes:

We have added Fig. R12C as Fig. 6F and Fig. R12A,B as Supplementary Fig. 7A,B and added the following to the Result section of the manuscript (Line 239 to 243):

“CUT&RUN for BRD9 and SS18 confirmed decreased binding of the ncBAF complex at the MYC promoter as well as RhOME2 and 3 loci (Fig. 6F). More generally, a genome-wide loss of binding of BRD9 and SS18 was observed after I-BRD9 treatment (Supplementary Fig. 7B, Supplementary Fig. 7C). Thus, confirming treatment-induced loss of ncBAF complex binding.”

Fig. R12: Chromatin occupancy of the SWI/SNF components before and after receiving I-BRD9 treatment. (A) The binding of BRD9 and SS18 at BRD9 binding site with and without I-BRD9 treatment; (B) Chromatin occupancy of active chromatin features and the SWI/SNF components (B) at the lost and gained open chromatin sites identified in the SMARCB1 reconstitution experiments and (C) at the MYC locus with and without iBRD9 treatment.

Other comments:

1. To better understand the sequencing samples, the authors should include a supplementary table about samples they generated, the sequencing depth, and quality of sequencing data (mapped reads etc.).

Changes:

We now included a table (Supplementary Table 1) summarizing this.

2. The level of reintroduced SMARCB1 in PDO should be shown to see its relative levels as compared to normal endogenous levels in cell lines that bear SMARCB1.

Answer:

Since the normal counterpart of MRT remains unknown, we compared *SMARCB1* expression levels in reconstituted MRT PDOs to endogenous *SMARCB1* expression levels in normal tissue-derived PDOs as well as PDOs derived from other, *SMARCB1* wildtype, pediatric kidney tumors. We found that the *SMARCB1* levels upon reconstitution in MRT PDOs are slightly elevated compared to the *SMARCB1* proficient PDOs (Fig. R13).

Fig. R13: TPM values of different patient-derived organoids from normal kidney or a pediatric kidney tumor entity.

Changes:

We have included Fig. R13 as Supplementary Fig. 1A.

3. Fig.1C only showed the changes of CTCF and RAD21 in the regions with ATAC-Seq changes, if the authors analyze the ChIP-seq of CTCF and RAD21 directly, how many peaks in total are altered?

Answer:

The vast majority of CTCF binding sites are altered (2-fold up or down: lost (CTRL specific) = 16,120, gained (SMARCB1 specific) = 83, stable = 2,212). ChIPing RAD21 in organoids appears to be challenging and the quality is not sufficient to perform a similar analysis.

4. Fig.1B, is the ATAC-Seq showing one of the replicate? Or merged data from replicates? The lost/gain at the bottom of Fig.1B, how are they called? The gained sites at the bottom only seem to show 4-5 peaks, but by eye many peaks in the top row of ATAC-Seq track seem to be gained peaks.

Answer:

The presented ATAC-seq data are the average of 3 independent experiments. The lost/gained sites are called using DEseq2 at FDR<0.05 and, at least a two-fold difference in accessibility between CTRL and SMARCB1+. All the peaks not fulfilling these criteria will not be called as “lost/gained” peaks. We now clarify this further in the figure legend of Fig. 1B.

We are grateful to the reviewer for pointing out this discrepancy to us and we apologize for this. There is indeed a mistake in the peak list of the original Fig. 1B. In this figure, there are two peaks from both lost and gained coordinates actually from other chromosomes, due to a bug in our original analysis script (Fig. R14). We corrected this in the new Figure:

Fig. R14: The updated lost and gained open chromatin regions in Fig. 1B.

Changes: We have updated Fig. 1B in the manuscript.

5. For Fig.1B, gene tracks should be added to better know are these noncoding regions or coding genes.

Answer:

We have added the gene tracks to Fig. 1B as suggested (Fig. R15) and included it in the revised version of our manuscript. The finalized figure is below and will be updated in the manuscript. The gained region does not have any genes or non-coding RNAs to add to the figure.

Fig. R15: The updated Fig. 1B.

6. Fig.2A, can RhOME3 also be included in the Hi-C map? Was the loop (circled) out only called in P103 PDO and won't be called in re-expression? Or is it quantitatively altered?

Answer:

Well taken. We included RhOME3 in the Hi-C map (Fig. R8, included in the revised manuscript as Fig. 2A). The circled loops are the loops identified in either Control or Control and SMARCB1+ PDOs. The dashed circles indicate that the loops are decreased after SMARCB1 reconstitution. Since the loop calling is also affected by local background, we consider quantitative difference a better indicator for measuring loop changes.

Changes:

We updated Fig. 2A,B in our manuscript including all the three loops (RhOME1-3).

7. Only 29 ATAC-seq peaks were consistently seen in all 3 PDOs, what are the genes associated with these? Is MYC loci one of the few with consistent changes of ATAC-Seq in all 3 PDOs?

Answer:

No, the *MYC* promoter signal only passes 2-fold threshold in P60. We think it is a bit mis-leading to use this cut-off, therefore we decided to remove the venn diagram.

Changes:

As mentioned above (Major concerns, Question 5C), we now decide to remove the Venn diagram and use the more quantitative k-means clustering analysis.

Minor:

Line 85 – references were not converted to numbered citations.

Changes:

The reference was changed to proper formatting.

Reviewer #4, expertise in single cell multi-omics (Remarks to the Author):

The authors used datasets from various experiments including single cell multi-omics data to study the effect of SMARCB1 mutation in malignant rhabdoid tumor (MRT). I suggest that the authors address the following comments:

1. A number of experiments were performed and it would be helpful to have a table to organize the experiments and datasets, including information on technology (RNA-seq, ATAC-seq, CHIP-seq, Hi-C, qPCR, 10x multiomics, etc), sample (P78, P60, P103, P156, ...), condition (control or SMARCB1+), and figures that is related to each dataset.

Changes:

We have included a table (Supplementary Table 1) summarizing the performed techniques and samples to give a comprehensive overview of the datasets at hand.

2. From single cell multi-omics data, the authors found that tumor cells are different from different patients and normal cells are less variable across patients. More analysis need to be done to dive into this.

First, the UMAP shown in Fig. 5C was plotted with both the scRNA-seq modality and the scATAC-seq modality. One can plot the two modalities separately to investigate whether the same pattern holds for each modality.

Answer:

Following the reviewer's suggestion, we plotted the two modalities separately and together and annotated the normal cells as well as different patient tumors (Fig. R16, only included as a response to the reviewers). Similar patterning is seen for both modalities plotted individually, comparable to the joined UMAP depiction.

Fig. R16: UMAPs of single modalities or combined modalities plotted as normal cells or patient ID (PCA: 1:25, LSI: 2:30).

Second, the authors found that enhancers RhOME1, RhOME2, and RhOME3 have different accessibility across patients. The following analysis can be performed to investigate what else cause differences between patient tumors: (1) using the scRNA-seq modality, one can perform differential expression analysis to find genes that are differentially expressed in the tumor cells between patients; (2) using the scATAC-seq modality, one can also perform differential accessibility analysis to find which regions are differentially accessible across patient tumors.

Answer:

Thank you for these suggestions. Following these, we performed differential gene expression and differential peak analysis comparing one patient tumor versus all other cells (including normal cells from all patients). We observed high variability in the number of genes and peaks called per patient, but we could not identify any underlying biological processes linking them to a role in tumorigenesis (Table R1, Fig. R17, only included as a response to the reviewers). Little overlap was observed between differential genes/ peaks called in each patient individually (Fig. R17). Gene ontology (GO) analysis on the differential genes of each patient sample (Fig. R17C), only revealed very general processes and no specific patterns can be identified, suggesting that these tumors share many of their tumor-driving processes. To perform the same analysis on the differential peaks, we first filtered for promoter region annotated peaks and secondly called closest protein-coding genes to these regions (Table R1). Comparable to the GO-analysis on the differential genes, the enriched terms are rather general, and no major differences of tumor-driving processes could be identified.

Table R1: Overview of differential genes, peaks, promoter peaks and associated genes by promoter peaks (protein coding only) called between tumor entities compared to all other cells (including normal cells).

Tumor	Differential genes	Differential peaks	Differential promoter annotated peaks	Associated genes by promoter peaks
P052	215	331	256	240
P041	72	268	139	124
P116	67	3	3	3
P138	83	937	569	530
P156	230	464	198	173

P166	138	2067	1158	1022
P168	41	23	18	17

Fig. R17: (A) Heatmap of differential genes of each patient sample ($\log_{2}FC > 1$, minimum percentage 10%). (B) Upset plot depicting overlap between the differential genes. (C) Top 10 GO-terms (biological processes) based on differential genes identified in each patient tumor. (D) Heatmap of differential peak of each patient sample ($\log_{2}FC > 1$, minimum percentage 10%). (E) Upset plot depicting overlap between the differential peaks. (F) Top 10 GO-terms (biological processes) based on closest protein-coding genes to promoter annotated peaks identified in each patient tumor.

3. In line 435, please give full name for SCT and TF-IDF, and specify which package and which version was used to perform the normalization.

We have added this information to the Methods section of the revised manuscript as well as included package versions.

Reviewers' Comments:

Reviewer #1:

Remarks to the Author:

The revised manuscript by Liu et al. entitled "SMARCB1 loss activates patient-specific distal MYC enhancers that drive malignant rhabdoid tumor growth" has addressed my comments to the original submitted manuscript. The additions have provided increased clarity and rigor. I appreciate the authors' thoughtfulness in the rebuttal and thoroughness in the changes in the manuscript. The study provides novel insights into the epigenetic regulation of MRT and could be of broader relevance to other tumors with SMARCB1 deletion or other BAF complex perturbations.

Reviewer #2:

Remarks to the Author:

The authors have addressed my major concerns.

Reviewer #3:

Remarks to the Author:

In the revised version, improvements were made, mostly by adding ChIP-seq of SMARCB1, Cut&Run of BRD9 and SS18, and additional BRD9 inhibitor data. Text edits were made to tune down some conclusions, including the title.

While I appreciate the revision, however, my honest opinion is that the central point/message this paper can deliver is limited; the novelty is not very strong (I do acknowledge PDO as an advanced model); it is unclear if any epigenome or chromatin topology changes observed are due to direct or indirect, or secondary effects of SMARCB1 loss (and compared to strong changes of CTCF binding, Hi-C surprisingly shows very little change); the mechanisms underlying MYC chromatin topology changes are unclear; the idea that cancer patients have patient-specific MYC enhancers is not particularly novel, and needs functional data to support that patient-specific enhancers are truly functional for MYC gene activation or MRT growth in a patient-specific manner; the treatment by iBRD9 while important is not particularly new. The authors generated multiomics datasets (I appreciate the efforts), but the conclusion from it is very limited.

1. About novelty issue: the authors responded that "The fact that we use MRT organoid models from different patients in our study allowed us to describe for the first time that MRT development can be driven by patient-specific epigenetic reprogramming caused by SMARCB1 loss". While this reviewer understands the difficulty to acquire a large number of MRT tissues, and appreciate the value of PDO as a more relevant model to study MRT than cell lines, but fairly speaking, data from this work is far from sufficient to conclude that "MRT development can be driven by patient-specific epigenetic reprogramming". The data so far can only support that there are putative enhancers (include several in the MYC locus) that appear different in different patients. Actually, the correlative data did not even prove that patient-specific MYC enhancers truly regulate MYC expression in a patient-specific manner (not to mention "drive tumor growth"). For example, by CRISPRi, does RhOME1 not regulate MYC expression in P103 or P78, but only RhOME2 does?

Title: "SMARCB1 loss activates patient-specific distal MYC enhancers that drive malignant rhabdoid tumor growth":

- Similar to above, there is no data in the paper can support this important conclusion. The paper identified some patient-specific PUTATIVE enhancers near MYC locus, but did not provide functional data to demonstrate any of these enhancers can regulate MYC gene expression, nor is there data that indicates any of these enhancers can drive MRT growth.

- "Drive" is a very strong word and should be tuned down. Something like "SMARCB1 loss activates patient-specific distal MYC enhancers in malignant rhabdoid tumor" is more supported by the data.

2. Disconnection between meta-analysis and MYC locus.

The paper's title is strongly focused on MYC locus, yet, the authors spend quite a lot of efforts on meta-analysis in Figs.1,4 built on ATAC-Seq changes with/without SMARCB1. In other figures, they solely focused on MYC enhancers to link to epigenetic/enhancer changes to this key oncogene expression and MRT growth. However, after reading the whole revised paper, this reviewer is confused if MYC enhancers (RhOME1,2,3) and promoter belong to the lost ATAC-Seq (n=1,211, Fig.2B,C) group or not? do they belong to K1/K2/K3 in Figure 4? Are MYC enhancers examples of meta-analysis? Or simply just because MYC locus being clinically important and shows Hi-C changes and thus becomes the focus?

3. Conceptually, the paper did not clearly demonstrate how SMARCB1 works in reprogramming the BRD9/SS18 landscapes of MRT. Comparing the 1,211 (lost) and 7,941 gained ATAC-Seq sites in Sup Fig.2B, a dominant pattern is that the gained sites are mostly bound by all three SMARCB1/BRD9/SS18 after SMARCB1 re-expression. In contrast, the lost sites are losing BRD9/SS18, but are only weakly bound by SMARCB1. These suggest that for the sites losing BRD9/SS18, SMARCB1 often does NOT directly inhibit them (as it does not even bind these sites). The authors need to consider a model that SMARCB1 re-expression recruited BRD9 and SS18 to the gained sites (7,941), so causing re-distribution of BRD9/SS18 from the ~1,211 sites. A model figure explaining how the authors interpret the role of SMARCB1 in organizing enhancers and BRD9/SS18 binding will be helpful (can be in sup figures). And do MYC enhancers follow this model or not? Can MYC locus be used as a typical example to illustrate how SMARCB1 works to reprogram BRD9/SS18 and to activate oncogenes for MRT growth?

4. Are the lost ATAC-Seq sites (Fig.2C) often associated with reduced gene expression? Gained sites associated with upregulation of nearby genes?

5. The new data of iBRD9 treatment followed by BRD9/SS18 ChIP-seq is helpful (Sup Figure 7B). If iBRD9 inhibits MRT growth (or causes differentiation) via shared mechanisms/targets as SMARCB1 restoration, then some targets should be common in Sup Fig.7B versus Sup Fig. 2B. There are in total 13,939 peaks of BRD9, how many BRD9 sites are weakened/lost after SMARCB1 re-expression, and how many were weakened by iBRD9, how many are shared? Similarly, how many SS18 sites weakened? Can these sites explain the phenotype of tumor growth inhibition after iBRD9 and SMARCB1 re-expression?

6. Fig.4A, what this figure's main message is? And why it should be in main Figure 4A. The figure shows that K1/K2 of ATAC-Seq sites lost in P103 can largely be seen in P60 and P78, but the K3 seems to be P103 specific. K3 seems to be the strongest enhancers in P103 if judged by K27ac level (Fig.4A, second most right column). So do the authors want to suggest that in each PDO, SMARCB1 can suppress a patient specific group of enhancers that tend to be highly active enhancers (super-enhancers, based on Sup Fig.5A,B). But then what do MYC enhancers belong to? is MYC a patient-invariant locus in all PDO models? Do RhOME1,2,3 of MYC belong to either K1/K2/K3 in P103? If they belong to K3, then how can they be invariant among patients? If they belong to K1/K2, then they are not the strongest enhancers? then what is the point of showing Fig.4A? and also what enhancers can be examples of K3 in P103? The message seems very unclear in this Reviewer's opinion.

7. Fig.5D, MYC gene/promoter does not show higher ATAC-Seq signals in at least three patients' tumor cells than in normal cells? Is MYC gene expression higher in tumor cells than in normal cells in the scRNA-seq data? Or should the normal cells be separated into every patient's normal cells to be compared to their respective tumor cells? All the RhOME1/2/3 are not very open in Fig.5D in normal cells yet MYC gene is still active, are there normal-cell-specific enhancers around that are highly active

in normal cells only?

Others:

The new data of SMARCB1 ChIP-seq is helpful. However, the data quality and analysis need clarification. Visual inspection of the ChIP-seq in Figure 2B finds SMARCB1 signals to be quite noisy. How many ChIP-seq peaks of SMARCB1 can be called from the P103 ChIP-Seq? Is it similar to previous ChIP-seq datasets of SMARCB1 in other papers and in MRT cell lines? Or is it noisier due to PDO model?

Line 266: "Here, we identify three putative super enhancers looping to the MYC promoter in MRT, as a consequence of SMARCB1 loss." This is not correct. The looping of MYC promoter with enhancers does exist in SMARCB1+ cells. So their presence is not a consequence of SMARCB1 loss, although SMARCB1 loss may impact the enhancers' activity. Indeed, in many other cases of normal or other cancer types bearing WT SMARCB1 (in literature), MYC often loops with one or some of these enhancers.

Thanks for generating the rebuttal figure, Fig. R10 (p(s) curve). I will suggest this to be added to Sup Fig. 3A.

In the rebuttal, the authors mentioned 16,120 CTCF (>80% of total) sites showed reduced binding after SMARCB1 re-expression. Did CTCF protein level change? How many replicates of CTCF ChIP-seq were done? It is surprising that this dramatic CTCF change did not accompany dramatic Hi-C loop/TAD change.

Did iBRD9 reduce BRD9/SS18 protein levels?

Can gene tracks be added to Fig.4B?

Reviewer #4:

Remarks to the Author:

The reviewer would like to thank the authors for their efforts on addressing the reviewer's previous comments.

The results on DE genes and differentially accessible regions partly support the observation that tumor cells are highly patient specific, as the DE genes highly expressed in each patient have little overlap. If the authors like, they can include this results. However, it is not clear why the GO terms corresponding to the DE genes are not specific and this is subject to future studies.

This reviewer does not have further concerns.

NCOMMS-23-03210-T-Rebuttal-#2: “SMARCB1 loss activates patient-specific distal MYC enhancers that drive malignant rhabdoid tumor growth”

We thank the reviewers for their positive response to our extensive revisions and are delighted to hear they are supportive of publication of our manuscript.

Reviewer #1 (Remarks to the Author):

The revised manuscript by Liu et al. entitled “SMARCB1 loss activates patient-specific distal MYC enhancers that drive malignant rhabdoid tumor growth” has addressed my comments to the original submitted manuscript. The additions have provided increased clarity and rigor. I appreciate the authors thoughtfulness in the rebuttal and thoroughness in the changes in the manuscript. The study provides novel insights into the epigenetic regulation of MRT and could be of broader relevance to other tumors with SMARCB1 deletion or other BAF complex perturbations.

Reviewer #2 (Remarks to the Author):

The authors have addressed my major concerns.

Reviewer #4 (Remarks to the Author):

The reviewer would like to thank the authors for their efforts on addressing the reviewer’s previous comments.

Reviewer #3 (Remarks to the Author):

In the revised version, improvements were made, mostly by adding CHIP-seq of SMARCB1, Cut&Run of BRD9 and SS18, and additional BRD9 inhibitor data. Text edits were made to tune down some conclusions, including the title.

While I appreciate the revision, however, my honest opinion is that the central point/message this paper can deliver is limited; the novelty is not very strong (I do acknowledge PDO as an advanced model); it is unclear if any epigenome or chromatin topology changes observed are due to direct or indirect, or secondary effects of SMARCB1 loss (and compared to strong changes of CTCF binding, Hi-C surprisingly shows very little change); the mechanisms underlying MYC chromatin topology changes are unclear; the idea that cancer patients have patient specific MYC enhancers is not particularly novel, and needs functional data to support that patient specific enhancers are truly functional for MYC gene activation or MRT growth in a patient specific manner; the treatment by iBRD9 while important is not particularly new. The authors generated multiomics datasets (I appreciate the efforts), but the conclusion from it is very limited.

Answer:

We would like to thank the reviewer for his/her critical evaluation of our manuscript. The main point of criticism brought up by the reviewer is the perceived lack of novelty. Unfortunately, we were unable to find what literature is being referred to, which makes it difficult to precisely gauge what the reviewer is referring to. Notably, the other three reviewers acknowledge the novelty of our findings and are supportive of publication in Nature Communications. We would like to reiterate that our paper contains the following novel observations:

1. In this paper, we describe for the first time intertumoral epigenetic heterogeneity on the level of the regulatory landscape. We use the *MYC* enhancer landscape, being a bona fide

- oncogene, as a showcase to describe intertumoral epigenetic heterogeneity in patients. Furthermore, we identify a novel putative *MYC* enhancer (RhOME2).
2. Our HiC and 4C data is one of the first chromosome conformation capture datasets generated on patient-derived organoid models allowing us to find patient-specific chromosome looping. No previous Hi-C data from MRT cells show the changes of chromatin interactions at the loci of well-known oncogenes.
 3. We are the first to generate a parallel single cell readout of transcriptome and chromatin accessibility (scMultiome) of MRT patient samples, representing the largest cohort for this rare tumor entity.
 4. To our knowledge, I-BRD9 treatment is only investigated in MRT cell lines. These cell lines do not recapitulate the molecular characteristics of the patients' tumor. The use of patient-derived organoids increases the translational benefit of our study, showing the efficiency of I-BRD9 to induce differentiation in MRTs like *SMARCB1* reconstitution. Our comparison of *SMARCB1* re-expression and I-BRD9 treatment sheds light on the function of I-BRD9 as a potential (maturation) therapy agent.

1. About novelty issue: the authors responded that “The fact that we use MRT organoid models from different patients in our study allowed us to describe for the first time that MRT development can be driven by patient-specific epigenetic reprogramming caused by *SMARCB1* loss”. While this reviewer understands the difficulty to acquire a large number of MRT tissues, and appreciate the value of PDO as a more relevant model to study MRT than cell lines, but fairly speaking, data from this work is far from sufficient to conclude that “MRT development can be driven by patient specific epigenetic reprogramming”. The data so far can only support that there are putative enhancers (include several in the *MYC* locus) that appear different in different patients. Actually, the correlative data did not even prove that patient specific *MYC* enhancers truly regulate *MYC* expression in a patient specific manner (not to mention “drive tumor growth”). For example, by CRISPRi, does RhOME1 not regulate *MYC* expression in P103 or P78, but only RhOME2 does?

Answer:

Thank you. We believe this comment is based on a misunderstanding. Our data demonstrate that PDOs that have been derived from MRT of different patients show a high degree of patient specificity (i.e, intertumoral heterogeneity) on the level of their regulatory landscape (as assessed by ATAC-seq). Since *SMARCB1* is the only recurrent mutation in these tumors, we assume that the heterogeneity is a consequence of the loss of *SMARCB1*. This heterogeneity is subsequently confirmed in several patient MRT tissues using scMultiome. However, we do agree with the reviewer that it would be more correct to state that MRT development is a consequence of changes in the regulatory landscape caused by *SMARCB1* loss and that these changes happen to be patient-specific.

Furthermore, all three RhOMEs have a high level of H3K27Ac in the different PDOs. Being a marker of active enhancers (Creighton et al., 2010), we can be certain that these regions are putative active super enhancers. Importantly, while RhOME2 has not been identified before as a putative *MYC* enhancer, RhOME1 and RhOME3 have been previously described as *MYC* regulating enhancers in particular in prostate cancer (RhOME1, putative: (Guo et al., 2021)) and leukemia (RhOME3, validated: (Bahr et al., 2018; Fulco et al., 2016)), but also in other cancers (Lancho & Herranz, 2018), as we described in our original manuscript (Line 164 - 166). We agree with the reviewer that further experimental work is required to confirm RhOME2 to be an active *MYC* enhancer. However, these experiments are not trivial as we would like to explain below.

Functional validation of these enhancers would involve demonstrating that RhOME inhibition results in decreased expression of *MYC*. This could, as suggested by the reviewer, be achieved through CRISPRi. We have spent a year optimizing the CRISPRi technique in our PDO models. We have tried multiple Cas9- and Cas12a-based CRISPRi systems (fused to KRAB or LSD1), but without success. This

could possibly be explained by the fact that the super enhancers stretch large genomic areas of up to 12kb making it difficult to target the entire enhancer. Multiple sgRNAs covering the entire genomic locus would be required for efficient enhancer inhibition. Transducing or transfecting multiple sgRNAs in parallel decreases the efficiency drastically of the necessary effect on enhancer inhibition. Second, organoids are renowned for the fact that they are difficult to transfect/transduce with plasmids containing large transgenes such as a CRISPR-based system. Additionally, stable expression using lentiviral transductions requires antibiotics selection of up to 2 weeks, during which cells having a growth disadvantage, due to for instance lower *MYC* expression, are rapidly outcompeted by cells in which the enhancer was not efficiently inhibited (and therefore have higher *MYC* expression and a competitive advantage). In line with this we also did not observe *MYC* downregulation upon CRISPR-mediated targeting of the *MYC* promoter itself. Continuing these experiments would result in an extensive optimization period, which in our opinion is beyond the scope of our current manuscript (especially because RhOME1 and RhOME3 are known *MYC* regulating enhancers, as mentioned above).

Again, we do acknowledge the importance of this issue, and we therefore address this pitfall in the Discussion (Line 271 to 272):

“Further mechanistic studies are required to further elucidate the super enhancer function in MRT development in more detail.”

Title: “SMARCB1 loss activates patient-specific distal *MYC* enhancers that drive malignant rhabdoid tumor growth”

- Similar to above, there is no data in the paper can support this important conclusion. The paper identified some patient specific PUTATIVE enhancers near *MYC* locus, but did not provide functional data to demonstrate any of these enhancers can regulate *MYC* gene expression, nor is there data that indicates any of these enhancers can drive MRT growth.
- “Drive” is a very strong word and should be tuned down. Something like “SMARCB1 loss activates patient-specific distal *MYC* enhancers in malignant rhabdoid tumor” is more supported by the data.

Answer:

We agree that choosing the phrasing ‘driving tumor growth’ indeed is not fully supported by the data. We decided to implement the reviewer’s suggestion and rephrase the title to:

*“SMARCB1 loss activates patient-specific distal *MYC* enhancers in malignant rhabdoid tumors”*

2. Disconnection between meta-analysis and *MYC* locus.

The paper’s title is strongly focused on *MYC* locus, yet, the authors spend quite a lot of efforts on meta-analysis in Figs.1,4 built on ATAC-Seq changes with/without SMARCB1. In other figures, they solely focused on *MYC* enhancers to link to epigenetic/enhancer changes to this key oncogene expression and MRT growth. However, after reading the whole revised paper, this reviewer is confused if *MYC* enhancers (RhOME1,2,3) and promoter belong to the lost ATAC-Seq (n=1,211, Fig.2B,C) group or not? do they belong to K1/K2/K3 in Figure 4? Are *MYC* enhancers examples of meta-analysis? Or simply just because *MYC* locus being clinically important and shows Hi-C changes and thus becomes the focus?

Answer:

We selected *MYC* as a showcase for intertumoral epigenetic heterogeneity focus, because our genome-wide analysis revealed that there were putative regulatory regions that were lost following

SMARCB1 reconstitution in the vicinity of this gene. In addition, our Hi-C analysis confirmed that these distal elements were interacting with the *MYC* promoter, an interaction that was lost upon SMARCB1 reconstitution. Because *MYC* is upregulated in the majority of cancers, including MRTs, we indeed decided to focus our analysis on this locus in particular. We already described this rationale in detail in our original manuscript (Line 129 to 133).

Furthermore, two of the putative *MYC* enhancers (RhOME 2 and 3) do indeed belong to the lost OCR sites and the K3 cluster (Fig. 2B,C; K3 cluster of Fig. 4A). The *MYC* promoter in P103 is not differentially accessible following SMARCB1 re-expression. It is not uncommon for *MYC* to also be expressed in normal cells, thus leading to an ATAC signal at the *MYC* promoter in the 'differentiated cells' after SMARCB1 re-constitution. Therefore, re-differentiating the MRT cells towards its normal counterpart does not automatically mean a complete abolishment of *MYC* activation in these cells. What does change, which we demonstrate also in our manuscript, is the way the *MYC* promoter gets activated, thus tumor-specific promoter-enhancer interactions.

Changes:

To further clarify our rationale to focus on the *MYC* locus, we now moved Supplementary Fig. 4B to the main Fig. 2C and describe it in the text as follows (Line 124-127):

“Ranking the most prominently lost loci upon SMARCB1 reconstitution, we found an interaction between the promoter of the MYC oncogene and a ~1.1 Mb distal region (Fig. 2A, C). This loop is the sixth most reduced interaction following SMARCB1 reconstitution, and the top ranked interaction involving a proto-oncogene”

3. Conceptually, the paper did not clearly demonstrate how SMARCB1 works in reprogramming the BRD9/SS18 landscapes of MRT. Comparing the 1,211 (lost) and 7,941 gained ATAC-Seq sites in Sup Fig.2B, a dominant pattern is that the gained sites are mostly bound by all three SMARCB1/BRD9/SS18 after SMARCB1 re-expression. In contrast, the lost sites are losing BRD9/SS18, but are only weakly bound by SMARCB1. These suggest that for the sites losing BRD9/SS18, SMARCB1 often does NOT directly inhibit them (as it does not even bind these sites). The authors need to consider a model that SMARCB1 re-expression recruited BRD9 and SS18 to the gained sites (7,941), so causing re-distribution of BRD9/SS18 from the ~1,211 sites. A model figure explaining how the authors interpret the role of SMARCB1 in organizing enhancers and BRD9/SS18 binding will be helpful (can be in sup figures). And do *MYC* enhancers follow this model or not? Can *MYC* locus be used as a typical example to illustrate how SMARCB1 works to reprogram BRD9/SS18 and to activate oncogenes for MRT growth?

Answer:

We share the reviewer's opinion that SMARCB1 itself does not directly cause the re-distribution of BRD9 or SS18, and that SMARCB1 re-expression most likely leads to re-distribution of the BRD9 and SS18 protein from the lost OCR sites to the gained OCR sites. We hypothesize that SMARCB1 re-expression leads to the re-formation of the cBAF complex, which in turn 'competes' with the ncBAF complex for binding at the gained OCR sites (Fig. R1). The *MYC* enhancer can only partially be used as an example for this model, as it reflects the lost site within our hypothesis, not the sites that are gained.

Fig. R1: Schematic model of the distribution of non-canonical (ncBAF) and canonical (cBAF) BAF complexes in SMARCB1 deficient (top) or proficient (bottom) cells. In the absence of SMARCB1, the BAF complex assembly is shifted towards the ncBAF composition leading to activated gene expression of, for instance, oncogenes. In SMARCB1 proficient cells, the cBAF complex competes with the ncBAF complex restoring the balance between the BAF complex compositions leading to re-distribution of the ncBAF complex and binding of the cBAF complex, thereby activating, for instance, expression of differentiation genes.

Changes:

We have added Fig. R1 to the manuscript as Supplementary Fig. 8 and refer to it in the Discussion (Line 301).

4. Are the lost ATAC-Seq sites (Fig.2C) often associated with reduced gene expression? Gained sites associated with upregulation of nearby genes?

Answer:

As suggested by the reviewer, we correlated all lost and gained OCR sites (Fig. 1C in the manuscript) with their putative gene targets using basal plus extension method in the rGREAT package (Figure R2, only included in the response to the reviewers). The lost and gained OCR sites are not strongly correlated with the expression of the nearby gene (very small fold changes). It is well-known that changes in chromatin accessibility are poorly correlated with gene expression even in single-factor perturbations (Kiani et al., 2022). This could be due to the challenge of assigning ATAC-seq peaks to their *bona fide* gene targets, or genes are regulated by both chromatin accessibility and other gene regulatory features such as chromatin interactions.

Figure R2: Expression heatmap of putative gene targets of lost and gained OCRs. The OCR-gene association was performed using basal plus extension methods using the rGreat package. The two RNA-seq replicates are shown individually in the heatmap. The boxplots give a summary of the genes per lost/ gained OCR and replicate. P-values were calculated using a Wilcoxon signed-rank test between the control and SMARCB1 samples using the average value of two replicates.

5. The new data of iBRD9 treatment followed by BRD9/SS18 ChIP-seq is helpful (Sup Figure 7B). If iBRD9 inhibits MRT growth (or causes differentiation) via shared mechanisms/targets as SMARCB1 restoration, then some targets should be common in Sup Fig.7B versus Sup Fig. 2B. There are in total 13,939 peaks of BRD9, how many BRD9 sites are weakened/lost after SMARCB1 re-expression, and how many were weakened by iBRD9, how many are shared? Similarly, how many SS18 sites weakened? Can these sites explain the phenotype of tumor growth inhibition after iBRD9 and SMARCB1 re-expression?

Answer:

In the Venn diagrams below, we show the numbers of BRD9 and SS18 peaks (Fig. R3, only included in the response to the reviewers' questions). As expected, only a minor increase in BRD9 and SS18 binding sites was observed after the treatments (i.e., SMARCB1 reconstitution or I-BRD9 treatment), whereas many binding sites were lost after treatment. For BRD9, 6,029 and 2,855 peaks remain unchanged after SMARCB1+ and I-BRD9, respectively. For SS18, 11,251 and 6,764 peaks remain unchanged after SMARCB1+ and I-BRD9, respectively. In conclusion, a high percentage overlap of decreased binding sites of BRD9 as well as SS18 can be seen in both treatments, suggesting a shared mechanism/ shared targets resulting in the similar phenotype seen.

Figure R3: The overlaps of the changed BRD9 and SS18 binding sites after SMARCB1 reconstitution (SMARCB1+) and BRD9 inhibition (I-BRD9).

6. Fig.4A, what this figure's main message is? And why it should be in main Figure 4A. The figure shows that K1/K2 of ATAC-Seq sites lost in P103 can largely be seen in P60 and P78, but the K3 seems to be P103 specific. K3 seems to be the strongest enhancers in P103 if judged by K27ac level (Fig.4A, second most right column). So do the authors want to suggest that in each PDO, SMARCB1 can suppress a patient specific group of enhancers that tend to be highly active enhancers (super-enhancers, based on Sup Fig.5A,B). But then what do MYC enhancers belong to? is MYC a patient-invariant locus in all PDO models? Do RhOME1,2,3 of MYC belong to either K1/K2/K3 in P103? If they belong to K3, then how can they be invariant among patients? If they belong to K1/K2, then they are not the strongest enhancers? then what is the point of showing Fig.4A? and also what enhancers can be examples of K3 in P103? The message seems very unclear in this Reviewer's opinion.

Answer:

The k-means clustering in Figure 4A was performed on the ATAC-seq data from all three PDOs, as well as the ChIP-seq data from P103. The main message we want to bring across is the patient-specificity seen for the K3 cluster, which is characterized by open chromatin (ATAC-seq), and high levels of H3K27Ac and RAD21 levels (ChIP-seq) in P103. RhOME 2 and 3, the two active MYC enhancers in P103, are part of the K3 cluster. However, as presented in the tornado plots (Fig. 4A), there are limited number of sites of the K3 cluster active in P60 and P78. All clustering methods, thus including k-means clustering, only categorize features following general trends. It does not mean that every single feature in a cluster will follow exactly the same pattern as the mean of this cluster.

The RhOMEs were the starting point to investigate all lost sites of the three PDOs leading us to this broader investigation of patient-specific lost enhancer sites. Thus, the RhOMEs can be used as an example of a patient-specific super-enhancers exemplified in the K3 cluster in P103.

Changes:

To explain this rationale in more detail, we added the following to the Result section (Line 180 to 183):

“Consistent with the function of super enhancers in control of cell identity³⁵, we identified the SOX protein motifs, including SOX2, SOX9 and SOX17, and functional annotations of many developmental processes that are associated with the K3 cluster-specific open chromatin sites, as well as RhOME2 and 3.1 OCRs (Supplementary Fig. 5C,D).”

7. Fig.5D, MYC gene/promoter does not show higher ATAC-Seq signals in at least three patients’ tumor cells than in normal cells? Is MYC gene expression higher in tumor cells than in normal cells in the scRNA-seq data? Or should the normal cells be separated into every patient’s normal cells to be compared to their respective tumor cells? All the RhOME1/2/3 are not very open in Fig.5D in normal cells yet MYC gene is still active, are there normal-cell-specific enhancers around that are highly active in normal cells only?

Answer:

It is not surprising that the normal cells within our dataset depict similar accessibility levels of the *MYC* promoter as the tumor cells (see Supplementary Fig. 6E). *MYC*, a transcription factor required for cell cycle progression and apoptosis under physiological conditions, is known to also be expressed in normal cells (Gnanaprakasam & Wang, 2017; Uhlén et al., 2015). Even though the *MYC* promoter depicts an ATAC signal in the normal cells, no ATAC signal is detected in the normal cells at the RhOME regions. Thus, this suggests that these super enhancer regions are, in the context of MRTs, tumor-specific. We do not preclude that in normal cells there are other *MYC* enhancers. The *MYC* enhancer landscape is a well-studied model in the context of cancer (Lancho & Herranz, 2018), but how this landscape is distributed in normal tissue remains to be explored.

Others:

The new data of SMARCB1 ChIP-seq is helpful. However, the data quality and analysis need clarification. Visual inspection of the ChIP-seq in Figure 2B finds SMARCB1 signals to be quite noisy. How many ChIP-seq peaks of SMARCB1 can be called from the P103 ChIP-Seq? Is it similar to previous ChIP-seq datasets of SMARCB1 in other papers and in MRT cell lines? Or is it noisier due to PDO model?

Answer:

Unfortunately, the reviewer does not indicate which published SMARCB1 dataset he/she is referring to. SMARCB1 is known to be a difficult protein to ChIP. To our knowledge, there is only one high-quality SMARCB1 ChIP-seq dataset published (Valencia et al., 2019). Our ChIP on SMARCB1 in PDOs does not yield the same level of quality as the ChIP experiment described by Valencia et al., which was performed in cell lines. It is extremely challenging to obtain a comparable amount of cell lysate (200-700ug) from PDOs (note that upon SMARCB1 re-expression the PDOs stop growing). With such high cell numbers, the basement membrane extract MRT PDOs are growing in represents a technical bottleneck in the CRISPR protocol as it interferes with the cross-linking procedure. We therefore had to perform the ChIP-seq for SMARCB1 with lower amount of cells, explaining the lower quality compared to the Valencia et al dataset. Nevertheless, the SMARCB1 ChIP-seq was of sufficient quality to perform quantitative analyses and to support our conclusions.

Line 266: “Here, we identify three putative super enhancers looping to the MYC promoter in MRT, as a consequence of SMARCB1 loss.” This is not correct. The looping of MYC promoter with enhancers does exist in SMARCB1+ cells. So their presence is not a consequence of SMARCB1 loss, although SMARCB1 loss may impact the enhancers’ activity. Indeed, in many other cases of normal or other

cancer types bearing WT SMARCB1 (in literature), MYC often loops with one or some of these enhancers.

Answer:

The reviewer is claiming that there is literature supporting loop formation of our identified enhancers to the *MYC* promoter in normal or other cancer cells. As mentioned in our original manuscript, loop formation with the *MYC* promoter has been described for RhOME1 and RhOME3 in other cancers, which is why we postulate in our manuscript that these enhancers are known to regulate *MYC* expression in other tumors and therefore highly likely to do so in MRT as well. However, RhOME2 has - to our knowledge - not been identified before and therefore represents a novel putative enhancer regulating *MYC* expression in, at least, MRT.

We do not claim that SMARCB1 or the SWI/SNF complexes are looping factors. More likely, these complexes modify the local chromatin states, which results in the recruitment of a looping factor such as cohesin. As shown in our previous publication (Liu et al., 2021), cohesin mediated enhancer-promoter loops can be highly context dependent. We hypothesize that upon loss of SMARCB1, the non-canonical BAF complex creates highly accessible regions maintaining the high level of cohesin binding at these super enhancers, which leads to increased looping between the *MYC* promoter and distal super enhancers.

Changes:

We have changed our wording in Line 266-268:

“Here, we identify three (putative) super enhancers involved in MYC regulation in MRT. SMARCB1 loss in these tumors leads to increased looping of these enhancers to the MYC promoter, likely activating its transcription. “

Thanks for generating the rebuttal figure, Fig. R10 (p(s) curve). I will suggest this to be added to Sup Fig. 3A.

Changes:

We have added Figure R10 as Supplementary Fig. 3C.

In the rebuttal, the authors mentioned 16,120 CTCF (>80% of total) sites showed reduced binding after SMARCB1 re-expression. Did CTCF protein level change? How many replicates of CTCF ChIP-seq were done? It is surprising that this dramatic CTCF change did not accompany dramatic Hi-C loop/TAD change.

Answer:

The protein levels of CTCF are reduced by about 50% upon SMARCB1 re-expression (Fig. R4, only included in the response to the reviewers' comment). Although it may seem counterintuitive that very little change in the Hi-C maps is observed, it is actually in line with published literature showing that a 50% reduction in CTCF protein levels does not lead to a dramatic change in TAD boundaries or loop formation. For instance, using degron experiments Nora et al. have shown that even an 85% reduction in CTCF levels does not lead to a massive change in 3D genome organization (Nora et al., 2017). However, the reduction seen in CTCF binding after SMARCB1 re-expression was one of the reasons for us to investigate the effect of this reduction on chromatin loop formation using HiC and 4C.

Fig. R4: Western blot of CTCF protein expression in MRT PDOs (P103) with and without SMARCB1 reconstitution. GAPDH was used as a loading control.

Did iBRD9 reduce BRD9/SS18 protein levels?

Answer:

Motivated by this comment, we performed a western blot to determine BRD9 and SS18 protein levels expression in MRT PDOs with and without 5 days of I-BRD9 treatment (which is identical to the treatment used for the ChIP-seq experiments) (Fig. R5). No decrease in BRD9 or SS18 protein levels can be observed upon I-BRD9 treatment, indicating that reduced BRD9 and SS18 binding at the *MYC* locus after I-BRD9 treatment, as measured by CUT&RUN, is not caused by a decrease in protein expression but rather due to reduced ncBAF complex binding.

Fig. R5: Western blot of BRD9 and SS18 after 5 days of I-BRD9 treatment [10 uM] in MRT PDOs (P103). GAPDH was used as a loading control. Size of protein ladder is depicted in the left side.

Changes:

We have added Figure R5 as Supplementary Fig. 7B and discuss it in the manuscript as following (Line 240 to 242):

“More generally, a genome-wide loss of binding of BRD9 and SS18 was observed after I-BRD9 treatment (Supplementary Fig. 7C,D), which was not caused by a treatment-induced decrease in BRD9 and SS18 protein expression (Supplementary Fig. 7B).”

Can gene tracks be added to Fig.4B?

Changes:

We have added the gene tracks to Fig. 4B.

Reviewers' Comments:

Reviewer #3:

Remarks to the Author:

I thank the authors for the work and their responses.